# Actionable human-water systems modeling under uncertainty

Laura Gil-García[1,2], Nazaret M. Montilla-López[3], Carlos Gutiérrez-Martín[3], Ángel Sánchez-Daniel[2], Pablo Saiz-Santiago[4], Josué M. Polanco-Martínez[1,5], Julio Pindado[1,6], Carlos Dionisio Pérez-Blanco[1,2]

[1]IME, Universidad de Salamanca, Francisco Tomás y Valiente, s/n, 37007, Salamanca Spain
[2]Department of Economic and Economic History, Universidad de Salamanca, Francisco Tomás y Valiente, s/n, 37007, Salamanca, Spain
[3]WEARE Research Group, Universidad de Córdoba, Córdoba, Spain
[4]Douro River Basin Authority. C/ Muro, 5, Valladolid, Spain
[5]Department of Statistics, Faculty of Sciences, Universidad de Salamanca, Plaza Merced s/n, 37008 Salamanca, Spain
[6]Business Administration Department, Universidad de Salamanca, Francisco Tomás y Valiente, s/n, 37007, Salamanca, Spain

*Correspondence to*: Laura Gil-García (lauragil_9@usal.es)

**Abstract.** This paper develops an actionable interdisciplinary model that quantifies and assesses uncertainties in water resources allocation under climate change. To achieve this objective, we develop an innovative socio-ecological grand

ensemble that combines climate, hydrological, and microeconomic ensemble experiments with a widely used Decision Support System for water resources planning and management. Each system is populated with multiple models (multi-model), which we use to evaluate the impacts of multiple climatic scenarios and policies (multi-scenario, multi-forcing) across systems, so as to identify plausible futures where water management policies meet or miss their objectives, and explore potential tipping points. The application of methods is exemplified through a study conducted in the Douro River Basin (DRB), an agricultural

basin located in central Spain. Our results show how marginal climate changes can trigger nonlinear water allocation changes in the Decision Support Systems (DSS) and/or; nonlinear adaptive responses of irrigators to water shortages. For example, while some irrigators barely experience economic losses (average profit and employment fall by <0.5%) under mild water allocation reductions of 5% or lower, profit and employment fall up to 12% (~24x more) where water allocation is reduced by 10% or less (~2x more). This substantiates the relevance of informing on the potential natural and socioeconomic impacts of

adaptation strategies, and related uncertainties, towards identifying robust decisions.

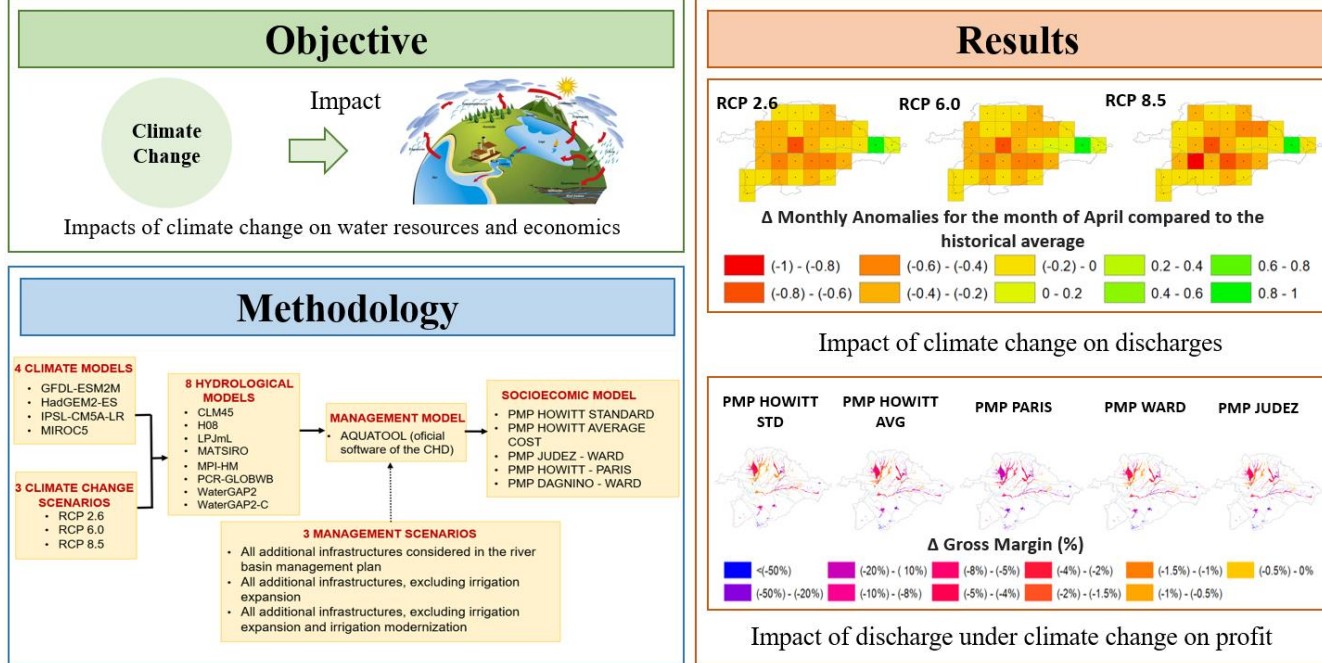

# 1 Introduction

Complex socioecological systems, including coupled human-water systems, are inherently difficult to manage (UNDRR, 2019). Periods of relative stability and predictability are interspersed with periods of unexpected, sometimes abrupt, change (UNDRR, 2021). These changes, even if small, can create ripples that cascade across systems and generate nontrivial environmental and socioeconomic impacts that are difficult to foresee—thus leading to uncertainty. We define uncertainty as a situation where "1) it is not possible to identify all plausible futures, or 2) assign a probability to each identified plausible future", which excludes probabilistic risk (Walker et al., 2003). Note that while point 2) refers to uncertainty in modeling that can be quantitatively assessed, point 1) cannot, and is accordingly not considered in our study (Knightian uncertainty [1](Knight, 1921)) Conventional consolidative modeling based on point predictions and optimization of expected performance risks

---

[1]Walker et al. (2003) identify different levels across the uncertainty spectrum: 1) determinism (where point predictions are reliable), 2) probabilistic risk (we know what plausible futures lie ahead of us as well as the associated probabilitie), 3) (deep) uncertainty type 1 (we do not know what inputs, parameters and/or model structures are right, nor their probability, but we can anticipate how the system will react to these), 4) (deep) uncertainty type 2 (we know we do not know), and 5) complete ignorance (we are not aware of what we do not know). Knightian uncertainty would fall in the levels 4-5, which precludes modeling. (Deep) uncertainty type 1 can be modeled, and has been modeled by model intercomparison projects such as (AGMIP, 2023; CMIP6, 2023; HEPEX, 2024; ISIMIP, 2023).

providing unrealistically precise information that gives a false appearance of uncertainty reduction (Hino and Hall, 2017). By ignoring (large parts of) the uncertainty inherent to modeling, the decision making processes informed by conventional consolidative models can miss their objectives, and under some conditions backfire and trigger crises (Anderies et al., 2006; Lempert, 2019). To allow humankind to embark on a sustainable and equitable development trajectory that delivers a satisfactory performance under multiple rather than a single plausible future (i.e., robust), a fundamental re-examination of the current approaches to modeling and uncertainty is necessary (IPCC, 2021). This calls for actionable forecasting methods that go beyond conventional consolidative models and point predictions and thoroughly quantify and assess uncertainty, so as to detect potential vulnerabilities to the system (including nonlinearities) and identify robust adaptation policies and trajectories (UNEP, 2021).

The literature identifies three fundamental sources of uncertainty in models: (1) input uncertainty arising from scenario design and data inputs (Marchau et al., 2019), (2) parameter uncertainty associated with the data and methods used to calibrate the model parameters (Tebaldi and Knutti, 2007), and (3) structural uncertainties associated with "the relationships between inputs and variables, among variables, and between variables and outputs" in models (Walker et al., 2003b). These uncertainties, which emerge within individual system models, can cascade across interconnected systems (UNDRR, 2019). Researchers have developed methods to quantify and assess scenario, parameter, and structural uncertainties in modeling, notably sensitivity analysis and multi-model ensemble experiments. Sensitivity analysis uses experiments representing the consequences of alternative sets of feasible assumptions (on scenario design, data, parameter values) to discover their implications (Groves et al., 2015; Lempert and Groves, 2010), while multi-model ensemble experiments group multiple models with alternative structures to produce a range of forecasts rather than a single point prediction (CMIP6, 2023; ISIMIP, 2023). Sensitivity analysis and multi-model ensembles can be combined into "grand ensembles" that quantify input, parameter, and structural uncertainties within a system through the ensemble spread (Athey et al., 2019). This approach has been used in disciplines such as climate sciences (e.g. Hagedorn et al., 2005; IPCC, 2014), economics (e.g. Krüger, 2017) and hydrology (e.g. Cloke et al., 2013). Recent research has combined grand ensembles over multiple ecological systems into multi-sector ensemble experiments such as the Coupled Model Intercomparison Project 6 (CMIP6) and the Inter-Sectoral Impact Model Intercomparison Project (ISIMIP) (CMIP6, 2023; ISIMIP, 2023). Echoing advances in socio-ecological research that have demonstrated the importance of considering the links between environmental change and human behavior when designing and assessing solutions to complex climate, water and other challenges (Pande and Sivapalan, 2017), multi-sector ensemble experiments have sought to incorporate human system aspects to their analyses. The conventional approach has been to exogenously model human systems through ensembles of macroeconomic models (typically Integrated Assessment Models), and subsequently transform these simulation outcomes into scenarios of greenhouse gas emissions that can be used to force ensembles of climate system models (Ferrari et al., 2022). Alternatively, human systems can be endogenously represented in the socioecological ensemble by explicitly representing the impacts of ecological systems on human behavior and responses, and vice versa, for example using microeconomic models (Sapino et al., 2022b). Finally, acknowledging that model

performance is conditional on its technical features but also on the modeling context and practices (Hamilton et al., 2019) research has given attention to Decision Support Systems (DSS) and studied its design (Guillaume, 2022), output assessment (notably via robust decision making, including multi-criteria evaluation (Groves et al., 2015; Maier et al., 2016; Marchau et al., 2019)), and output interpretation (including the study of beliefs and biases, path dependence, incentives, politics and power, information gaps and filtering, and others (Cook et al., 2018; Peters and Nagel, 2020; Quiggin, 2012)).

However, the application of these practices to human-water systems modeling, management and planning appears to be limited. In a recent literature review of uncertainty quantification in human-water system models (including DSS used to support management and planning) across 198 studies, it was found that most studies focused on partial assessments of input (148 of 198 studies) or parameter uncertainties (40) through local sensitivity analysis, while structural uncertainties (7) were typically neglected (González-López et al., 2023). Few studies quantified two sources of uncertainty (31), and none quantified

all three sources of uncertainty, which reflects on the nontrivial computational costs of conducting multi-model and sensitivity analyses across multiple systems. Notably, 51 studies included a DSS or water resources management model such as WEAP or MIKE, of which 35 accounted for input uncertainty, 5 for parameter uncertainty and 3 for structural uncertainty. Not a single study in the review, neither DSS nor other models, quantified uncertainties in both human and water systems (i.e., studies quantified uncertainties either in human or water systems). While integrated human-water systems models (including DSS)

abound in the literature (Baccour et al., 2022; Graveline, 2020; Gil-García et al., 2023; Li et al., 2020; Martínez-Dalmau et al., 2023; Pande and Sivapalan, 2017; Ward, 2021), and examples of model intercomparison experiments (e.g., HEPEX (2024)) and sensitivity analysis do exist (Puy et al., 2022; Saltelli, 2019) particularly in water systems, in practice water resources modeling (including DSS for planning and management) ignores uncertainties within and across water and/or human systems (OECD, 2021). This is also observed in the wider natural resources literature, where multi-system model-intercomparison

experiments to quantify structural uncertainties address ecological (and not human) systems (AGMIP, 2023; CMIP6, 2023; ISIMIP, 2023).

We argue that to develop water policies that are sensitive to climate change and other key sources of uncertainty, including the adaptive responses by human agents, it is necessary to deliver actionable interdisciplinary modeling that quantifies and assesses uncertainty. To achieve this objective, we propose an innovative human-water system grand ensemble that combines

climate, hydrological, and microeconomic ensemble experiments with a widely used DSS for water resources planning and management, named AQUATOOL (Andreu et al., 1991). The proposed modeling framework is illustrated with an application that quantifies structural uncertainties, as well as input uncertainties via climate change scenarios, although it can be expanded to quantify parameter and other input uncertainties (with nontrivial computational costs-see Section 5). In a first step, we use surface hydrology forecasts under alternative climate scenarios (RCP 2.6, RCP 6.0, RCP 8.5) obtained from ISIMIP to force

the DSS AQUATOOL, which yields information on water allocation to water users in the basin. In a second step, we assess the adaptive responses by irrigators to water allocation decisions, and their repercussions in terms of income, employment,

and water and land use changes. The grand ensemble adopts a modular approach where models at each system level operate independently in modules, which are subsequently interconnected through sets of protocols, i.e., rules designed to manage relationships among modules (Essenfelder et al., 2018). Each system is populated with multiple models (multi-model), which we use to evaluate the impacts of multiple climatic scenarios and policies (multi-scenario, multi-forcing) across systems. The uncertainty range provided through the ensemble spread can reveal relevant tradeoffs and vulnerabilities, including potential nonlinearities, thus providing valuable information that can be used to revise strategies and policies, including by adapting models to account for expert feedback, until a robust policy is agreed upon (Marchau et al., 2019). The methods are exemplified through an application to the Spanish share of the Douro River Basin (DRB).

## 2 Case study area: The Douro River Basin

The DRB in Spain covers an area of 78,889 km² and stretches over eight regions (NUTS2[2]), of which the Castile and León Region is the most relevant one (98.25% of the basin's total area). The region experiences an average annual rainfall of 450-500 mm, with lower figures in the central part, where most of the agricultural area is situated, and higher precipitation in the mountainous areas surrounding the basin. Low rainfall values are complemented in agriculture with an expanding irrigation supply that, albeit representing 10% of the total agricultural land, already claims 89% of the total water use of 4366 million m³/year in 2021—which is expected to increase to 4692 million m³/year and 4688 million m³/year by 2027 and 2033, respectively, mainly driven by irrigation expansion. Water supply has decreased on the other hand from 14231 million m³/year during 1940-2005 to 12777 million m³/year over 1980-2005, and although on average this is still sufficient to meet the growing demand, drought spells are increasing both in frequency and intensity (Field et al., 2014). Agriculture, the main user and the one generating the lowest market added value from water, suffers most of water allocation restrictions as per the Spanish use priority rules established in the Drought Management Plans (DRBA, 2018).

Agricultural lands represent more than a half of the Douro River Basin's total area (5.7 million ha) and include rainfed crops such as wheat (26%), barley (23%), rye (2%), sunflower (6%); and vineyard (2%). Irrigated crops include cereals such as maize (4%), alfalfa (2%), vegetables (1%) and sugar beet (1%). Surface water resources serve as the primary irrigation water source, representing on average 82% of the basin's water supply (DRBA, 2022). The relevant administrative unit for irrigation in the DRB (and in other regions in Spain) is the Agricultural Water Demand Units (AWDUs), which are also the agents in the microeconomic models.

---

[2] The European Union (EU) uses a system called NUTS (Nomenclature des Unités Territoriales Statistiques) to categorize its economic regions. In Spain, NUTS 2 level corresponds to regional divisions.(Eurostat, 2020)

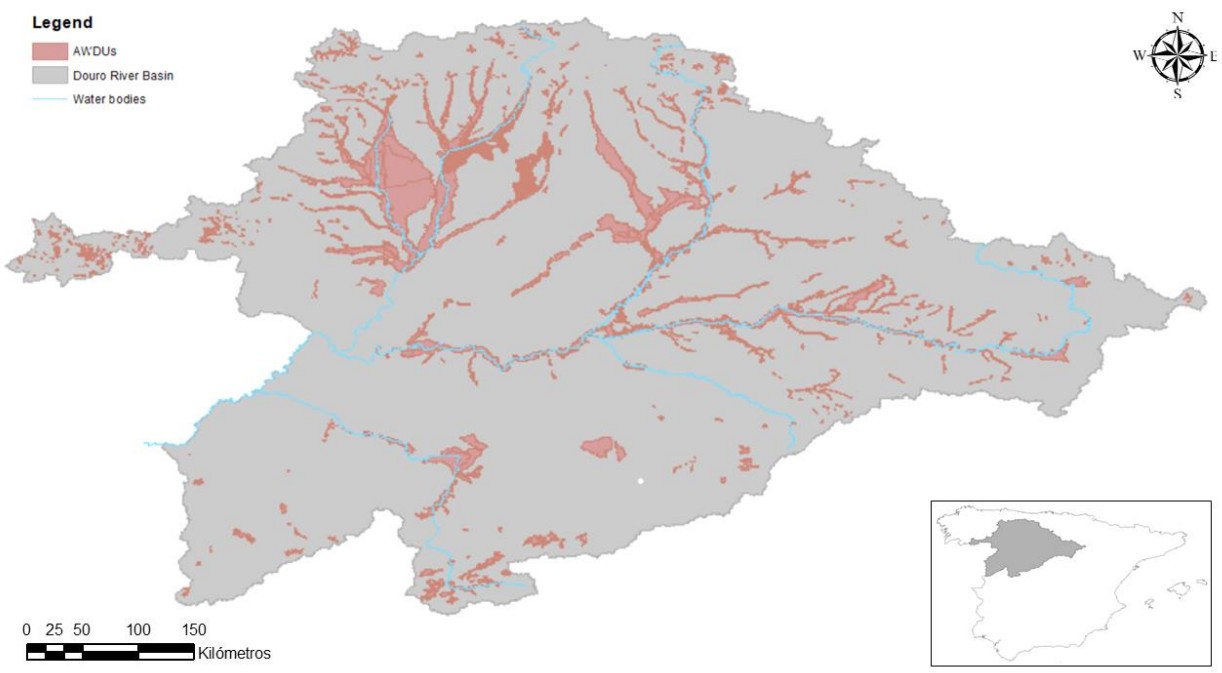

**Figure 1. Localization of DRB and detail of its AWDUs.**

**3 Methods: a modular hierarchy of socio-ecological ensembles**

We build a socio-ecological grand ensemble around AQUATOOL, a widely used DSS for watershed planning and management of water resources systems with applications to real planning cases in Spain, Ecuador, Brazil, Italy, Algeria, Mexico, Bosnia, Chile, Peru, Argentina, and Morocco, inter alia. The grand ensemble comprises four modules, each of which

representing a key system in the human-water conundrum: the climate system, modeled by ISIMIP (2023) through an ensemble of Global Circulation Models (GCMs); the water natural system, also modeled by ISIMIP (2023) through an ensemble of Global Hydrological Models (GHMs); the water management system, modeled through an ensemble comprising alternative setups of the DSS AQUATOOL; and the human system, modeled through an ensemble of microeconomic mathematical programming models. The coupling among modules is implemented in three steps (Fig. 2):

-Step 1 is done externally to our model by ISIMIP Protocol 2b (2023) and includes the simulation of discharge data by forcing the ensemble of GHMs with climate change forecasts obtained from the ensemble of GCMs under alternative climate scenarios;

-Step 2 imports discharge outputs from the GHMs ensemble into the AQUATOOL ensemble and produces data on water allocations under alternative water management scenarios; and

-Step 3 uses water allocation data to force an ensemble of microeconomic models that represent humans' behaviour and responses and simulate changes in land use, water use, income, and employment.

The upshot of this coupling process is a database of plausible futures that assesses the repercussions of climate change and adaptation scenarios on the water and human systems, while accounting for input (climate and management scenarios), structural and parameter uncertainties in modeling, as well as the cascading uncertainties across coupled ecological and human systems.

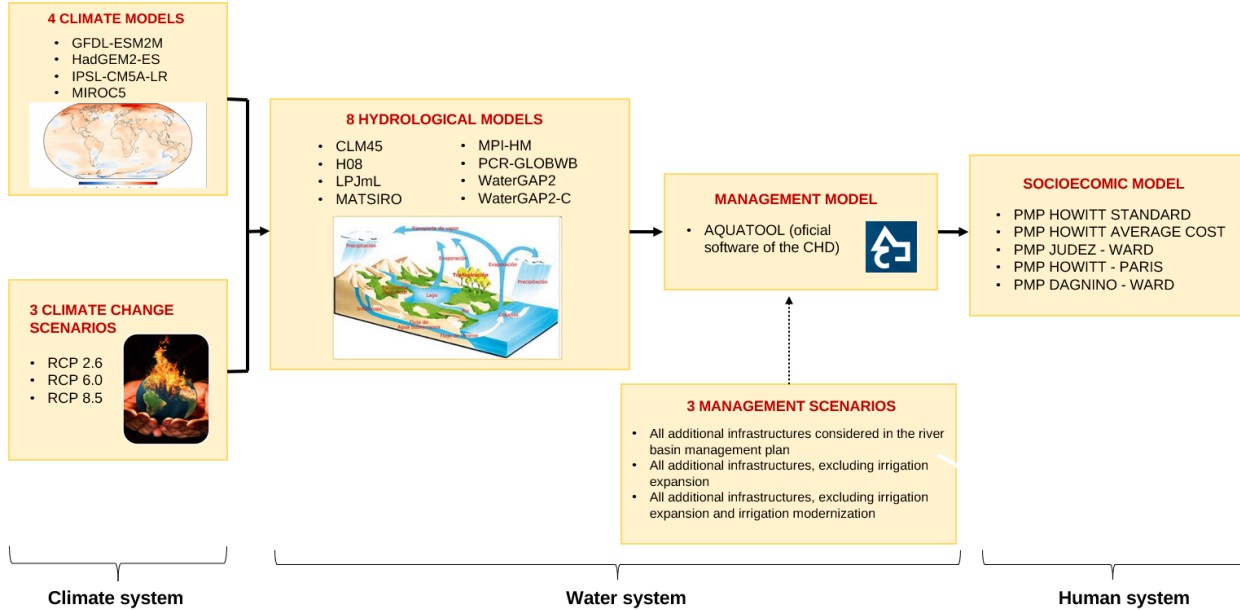

**Figure 2. Human-water system grand ensemble.**

The following subsections describe the components of the grand ensemble, namely the climate scenarios (Sect. 3.1); the management scenarios (Sect. 3.2); and the modules, including the climate system (Sect. 3.3), water natural system (Sect. 3.4), water management system (Sect. 3.5), and human system modules (Sect. 3.6).

### 3.1 Climate scenarios

The modeling exercise encompasses three of the original four RCP scenarios (ISIMIP Protocol 2b (2023) considers RCP2.6,RCP6.0, and RCP8.5, which outline trajectories used by the IPCC (2021, 2014) to depict various potential climate futures based on future greenhouse gas emissions. In our simulations, for all scenarios we assumed present day socio-economic conditions (2005 economic development, levels of population, land use and management consistent with management scenarios in AQUATOOL).

- RCP 2.6 outlines a climate scenario where CO2 emissions decline to zero by 2100, with methane emissions at 50% of 2020 levels and sulphur dioxide emission at 10% of 1980-1990 levels. Negative CO2 emissions (e.g., via tree CO2 sequestration) averaging 2 GtCO2/yr, are incorporated. This pathway aims to keep global temperature rise below 2ºC by 2100.
- RCP 6.0 foresees a peak in emissions by 2080, followed by a decline. It involves initially high greenhouse gas emissions, and stabilization of radiative forcing post-2100, leading to a projected 3-4ºC temperature rise with CO2 reaching 670 ppm.
- RCP 8.5 depicts a scenario where emissions keep increasing throughout the 21st century, which is typically regarded as unlikely but still possible. Initially viewed as a worst-case scenario with overestimated coal emissions, it continues to be employed to day for predicting mid-century and earlier emissions based on existing policies.

## 3.2 Management scenarios

The water management ensemble comprises three different setups of the AQUATOOL model. Each model setup corresponds to one alternative management scenario with specific developments of reservoirs, canals, irrigated land, and irrigation infrastructure. These management scenarios are the outcome of the public consultation process led by the basin authority and implemented during the third river basin planning cycle (2022-2027), which crystalized in the DRB Management Plan (DRBA, 2022). Under the management scenario 1 (M01), all the new developments proposed in the river basin plan are implemented; in the management scenario 2 (M02) all new developments proposed in the river basin plan, excluding irrigation expansion, are developed; and in the management scenario 3 (M03) all new developments proposed in the river basin plan, excluding irrigation expansion and irrigation modernization, are developed. The specific developments carried out under each management scenario are detailed in Appendix A.

## 3.3 Climate system module

Climate change forecasts are produced by the ISIMIP Protocol 2b (2023) by simulating the impacts of the three climate change scenarios (Sect. 3.1) using four GCMs. Each of the four GCMs is combined with each of the 3 RCP scenarios, thus generating 12 climate scenarios. The outputs from GCMs are used, in turn, to force GHMs (see next sub-section). The four GCMs include:

- GFDL-ESM2M combines atmospheric and oceanic circulation models, land dynamics, and biogeochemical processes like the carbon cycle. This model, a collaborative effort involving various institutions under the leadership of the Geophysical Fluid Dynamics Laboratory of the NOAA, aims to study climate and ecosystem interactions, both natural and human induced. It includes components for the atmosphere, land, and oceans, tracking factors such as aerosols, precipitation, and sea ice dynamics. The model also monitors chemical and ecological tracers that impact nutrient cycles, plant growth, and more. By integrating these components, the GFDL-ESM2M provides a comprehensive

understanding of how Earth's ecosystems interact with the climate system. For additional details and a mathematical statement, the reader is referred to Dunne et al. (2013, 2012).

-      HadGEM2-ES is part of the broader HadGEM2 model family involving diverse model setups that vary in complexity while sharing a unified physical structure. This version of the HadGEM is the second generation and includes, among other features, a well-resolved stratosphere. The HadGEM is developed in the Hadley Center, Met Office (UK) and is one the most well known full global climate models. For additional details and a mathematical statement, the reader is advised to Collins et al. (2011).

-      IPSL-CM5A-LR model is a comprehensive and full Earth System Model (ESM) and is developed in the Institut Pierre Simon Laplace (IPSL) (France). The model offers a versatile platform for addressing diverse scientific questions. It comprises two sets of physical models, including ocean extensions. The model's configurations can vary in terms of physical parameterizations, resolution, components (ranging from atmosphere and land to a full ESM), and processes (covering physical, chemical, aerosol, and carbon cycle processes). At its core, IPSL-CM5 integrates components for

the land surface, atmosphere sea ice, and ocean, along with biogeochemical processes, including stratospheric and tropospheric chemistry, aerosols, and terrestrial and oceanic carbon cycles. For additional details and a mathematical statement, the reader is advised to Dufresne et al. (2013).

     -      MIROC 5 (Model for Interdisciplinary Research on Climate version 5) is an Atmospheric and Oceanic GCM developed in the Atmosphere and Ocean Research Institute at University of Tokyo (Japan), MIROC is an advanced

climate model designed to better simulate the average climate, variability, and climate change resulting from human-induced radiative forcing. This model was tested through a century-long control experiment with specific atmospheric and oceanic resolutions, and its performance was compared to observations and a previous model version with varying spatial resolutions. For additional details and a mathematical statement of the model, the reader is referred to Watanabe et al. (2010).

## 3.4 Water natural system module

Water discharge forecasts are produced by the ISIMIP Protocol 2b (ISIMIP, 2023) forcing eight GHMs with the simulation outputs of the four GCMs. GHMs provide spatially aggregated information within standardized grids of 0.5°x0.5°. The eight GHMs include:

-      CLM4.5 explores the cycling of water, trace gases, chemical elements and energy. The model components include bio-geophysics, the hydrologic cycle, dynamic vegetation and biogeochemistry. The land surface is categorized into glacier, lake, wetland, urban, and vegetated areas, with further subdivisions for plant functional types. For additional details and a mathematical formulation, the reader is referred to Oleson et al. (2013).

- H08 represents a global hydrological model organized by grid cells, featuring six sub-models designed to explicitly replicate the interplay between the natural water cycle and human activities worldwide. The model maintains a nearly complete water balance. In 2016, water abstraction schemes were improved, and a groundwater scheme was added. CLM4.5 in ISIMP protocol 2b is run using inputs from the climate models: IPSL-CM5A-LR, HadGEM2-ES, GFDL-ESM2M, MIROC5. For additional details and a mathematical statement of the model, the reader is referred to Hanasaki et al.(2018).

- LPJmL is a model that focuses on water balance and irrigation processes, with the latest version distinguishing between different irrigation systems. It's designed to study the impact of replacing natural vegetation with agroecosystems due to rising $CO_2$ levels and climate change. Additionally, it plays a key role in assessing future ecosystem services, considering factors like climate, $CO_2$ levels, land management, and land use change. LPJmL in ISIMP protocol 2b is run using inputs from the climate models: IPSL-CM5A-LR, HadGEM2-ES, GFDL-ESM2M, MIROC5. For additional details and a mathematical statement of the model, the reader is referred to Bondeau et al. (2007).

- MATSIRO is meant to work with a climate system research model. It's used for climate studies covering various time scales and resolutions. MATSIRO focuses on representing essential land-atmosphere water and energy exchange processes in a physically based, yet straightforward manner, making it a valuable tool for climate research. MATSIRO in ISIMP protocol 2b is run using inputs from the climate models: IPSL-CM5A-LR, HadGEM2-ES, GFDL-ESM2M, MIROC5. For additional details and a mathematical formulation of this model, the reader is referred to Takata et al. (2003).

- MPI-HM is a model that focuses is solely on calculating water fluxes, excluding any considerations for energy balance calculations. MPI-HM is used for high-resolution river routing in hydrological research. MPI-HM in ISIMP protocol 2b is run using inputs from the climate models: IPSL-CM5A-LR, GFDL-ESM2M, MIROC5. For additional details and a mathematical statement, the reader is referred to Stacke and Hagemann (2012).

- PCR-GLOBWB is a model that simulates water dynamics for each grid cell on a daily basis. It tracks water storage in soil and groundwater layers, as well as exchanges like infiltration, percolation, and capillary rise. The model includes atmospheric interactions, such as rainfall and evapotranspiration, and connects water use in agriculture, industry, and households to daily hydrological processes. The simulated runoff and water flow are subsequently routed through river networks, interconnected with water allocation and reservoir operation schemes. PCR-GLOBWB in ISIMP protocol 2b is run using inputs from the climate models: IPSL-CM5A-LR, HadGEM2-ES, GFDL-ESM2M, MIROC5. For additional details and a mathematical statement, the reader is advised to van Beek and Bierkens (2009).

- WaterGAP2 & WaterGAP2-2C: are a global freshwater model that assesses water flows and storage across continents, factoring in human impact from water abstractions and dams. It helps analyze water scarcity, droughts, floods, and the influence of human actions on groundwater, wetlands, streamflow, and sea-level rise. The model relies on climate data, surface water information, land characteristics, and more for its inputs. WaterGAP2 & WaterGAP2-2C in ISIMP

protocol 2b is run using inputs from the climate models: IPSL-CM5A-LR, HadGEM2-ES, GFDL-ESM2M, MIROC5. For additional details and a mathematical statement of the model, the reader is referred to Alcamo et al. (2003).

### 3.5 Water management system module

AQUATOOL serves as a DSS designed for editing, implementing, reviewing, and analyzing hydrologic models, with a specific emphasis on integrated watershed management. It provides detailed data on the qualitative and quantitative condition of water bodies as well as water allocation, across both space and time. AQUATOOL is structured into various modules, each with its own or model or software. In our application to the DRB, we utilize the module termed "AQUATOOL" to execute a comprehensive longitudinal and spatial analysis of the impacts of climate change and discharge variations under alternative management scenarios on water bodies and water allocation. The impacts of climate change models and management scenarios on surface water bodies are simulated according to continuity or equilibrium principles, while both unicellular and multicellular models are used for groundwater bodies. The water allocation, on the other hand, is determined relying on a network optimization algorithm. The model is calibrated following a positive approach that aims at minimizing the difference between simulated and observed water allocations, observed discharges and reservoir levels (PUV, 2020).

The data inputs for setting up the AQUATOOL in the DRB are accessible online (Mírame-IDEDUERO, 2023), excluding the discharge series under natural conditions for the model baseline conditions (no climate change), which must be generated. To obtain these discharge series under natural conditions, daily precipitation series from 1940 to 2018 are processed using the EVALHID tool (Lerma et al., 2017) in addition to SIMPA (Sistema Integrado para la Modelación del proceso Precipitación Aportación) (CEDEX, 2020).

AQUATOOL is openly accessible for academics and practitioners, while private for profit companies have to pay a fee.

### 3.6 Human system module

The human system module is conformed by an ensemble of five Positive Mathematical Programming (PMP) models. PMP modeling also adapts a positive calibration approach capable of reproducing the choices of the reference year without error. PMP was first formalized by Howitt (1995) and has since been the dominant technique for calibrating mathematical decision-making models in the agricultural sector. In general, these models include a non-linear component within the objective function, which can be yield or cost. The original parameter, yield ($y_i$) or cost ($c_i$), is replaced by a crop area-dependent function ($c_i = \alpha_i + \frac{1}{2}\beta_i x_i$ or $y_i = B0_i + B1_i x_i$), so that when the area of a crop ($x_i$) expands, its yield decreases (its cost increases), and vice versa being $B0_i$, $B1_i$, $\alpha_i$ and $\beta_i$ the calibrating parameters (intercept and slope) for yield and cost linear functions.

Five alternative different PMP calibration techniques have been included into the human system ensemble, namely the standard approach of Howitt (1995), the average cost approach (Heckelei et al., 2000), and Paris (1988), Júdez et al. (2001) and Dagnino and Ward (2012) approaches, which we briefly introduced below. For a detailed description and mathematical statement of the model, the reader is referred to the original papers. While all of these approaches reproduce the reference/calibration year without error, the objective function and agents' responses during the simulations do differ, often significantly.

- Standard approach (Howitt, 1995). The original work included a yield function, which in this case has been replaced by a cost function. This method needs 2 stages to calibrate. First, the dual values ($\mu_i$) of some calibration constraints are obtained using a linear model. From these dual values, the observed cost ($cost_i^0$) and the observed area ($x_i^0$), the calibration coefficients of the cost function ($\alpha_i$ and $\beta_i$) are obtained. As noted by Heckelei et al. (2000), a key problem with the standard approach is the underdetermination of the calibration parameters.
- Average cost approach (Heckelei et al., 2000). The average cost approach is similar to Howitt (1995), but the calibration parameters are determined in such a way that, for the reference year, the value of the cost function coincides with the observed average cost.
- Paris (1988) eliminates the first calibration parameter.
- Judez et al. (2001) skip the first phase of Howitt's method and relies on external information to calibrate the model, land rent ($Land\ Rent$) and average income per crop ($Average\ Income_i$).
- Dagnino and Ward (2012). This method also skip the first phase and directly calibrates a yield function with the parameters $B0_i$ (intercept) and $B1_i$ (slope) from observed yield ($yield_i^0$), average income ($Average\ Income_i$) and price per crop ($price_i$).

**Table 1. PMP calibrating parameter by method**

| | Linear calibrating parameter | | Quadratic calibrating parameter |
|---|---|---|---|
| Standard approach | $\alpha_i = cost_i^0$ | | $\beta_i = \dfrac{\mu_i}{x_i^0}$ |
| Average cost approach | $\alpha_i = cost_i^0 - \mu_i$ | | $\beta_i = \dfrac{2\mu_i}{x_i^0}$ |
| Paris approach **(1988)** | $\alpha_i = 0$ | | $\beta_i = \dfrac{cost_i^0 + \mu_i}{x_i^0}$ |
| Judez et al. **(2001)** | $\alpha_i = cost_i^0 - \dfrac{1}{2}\beta_i * x_i^0$ | $\beta_i = \dfrac{2 * AverageIncome_i - LandRent}{x_i^0}$ | |
| Dagnino and Ward **(2012)** | $B0_i = yield_i^0 - B1_i * x_i^0$ | | $B1_i = \dfrac{-AverageIncome_i}{price_i * x_i^0}$ |

All models maximize a quadratic objective function where the only relevant attribute is profit, as measured by the profit, subject only to soil and water constraints. The data used for the calibration of the five PMP models is available in Appendix B.

## 4 Results

We conduct a set of simulations in three steps, following the hierarchy detailed in Fig. 2. Step 1, which is performed externally to our model by ISIMIP (2021) Protocol 2b, produces discharge data by forcing 8 GHMs with climate change forecasts produced by an ensemble of 4 GCMs under 3 climate change scenarios. This results in 86 plausible futures (note that not all GHMs can run simulations using the outputs produced by the GCMs, as explained in Sect. 3.4). Discharge data is produced in regular grids of 0.5ºx0.5º and transformed into discharge anomalies (%) by comparing GHMs forecasts under climate change (2006-2040 and 2006-2070 periods) to simulations using historical data (45 years in historical series from 1961 to 2005). Figure 3 illustrates longitudinal discharge anomalies for a critical section of the basin in the border between Portugal and Spain, using a 12-month moving average Most combinations of models and scenarios in Fig. 3 forecast a reduction in discharge. Discharge reductions exhibit a more significant impact during the 2040-2070 period compared to the earlier 2006-2040 period, exacerbated by the peak in greenhouse gas concentrations and the worsening effects of climate change on the water cycle.

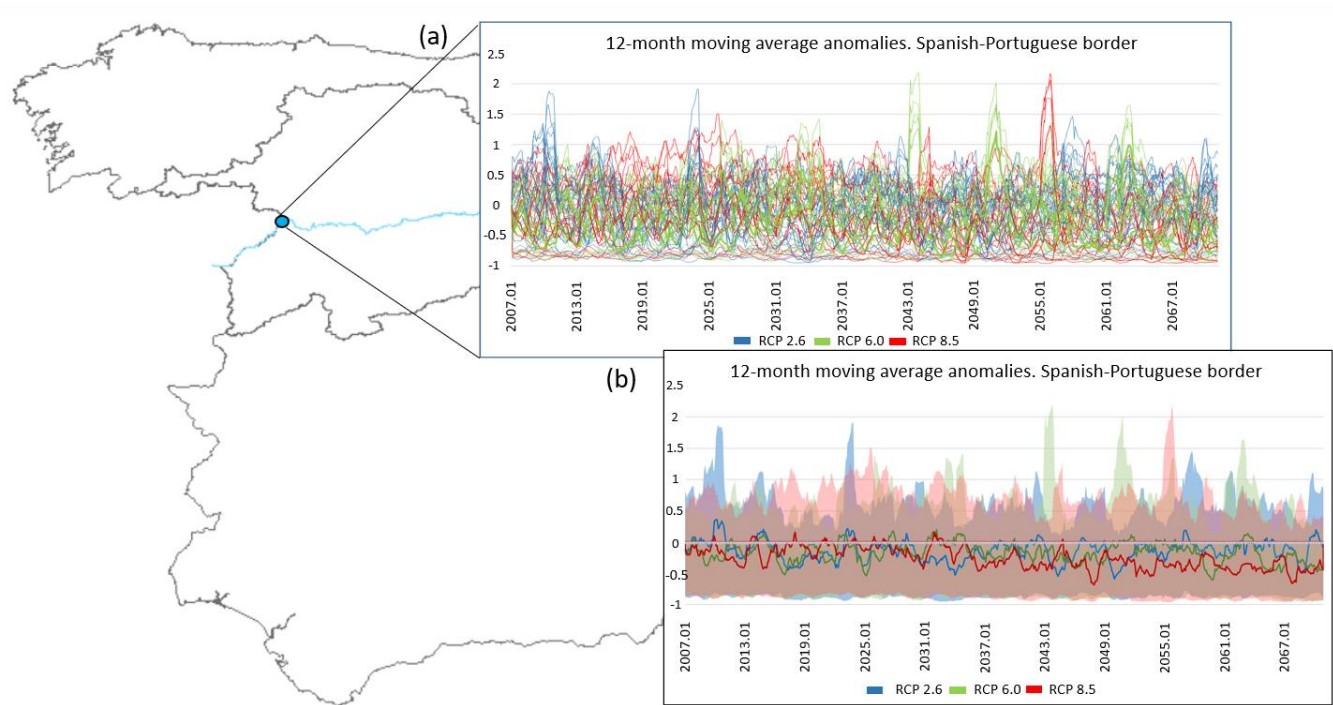

**Figure 3. (a) Longitudinal discharge anomalies in the border between Portugal and Spain (moving average of 12 months). (b) Ensemble spread and best estimate of discharge anomalies for the border between Portugal and Spain, moving average of 12 months.**

In Step 2, anomalies in discharge reported by GHMs are imported into the water management system ensemble to obtain longitudinal series of water allocation for each AWDU under 3 alternative management scenarios (see Sect. 3.2). To this end, we follow the approach by MAGRAMA (2017) to adjust the discharge series under natural conditions in AQUATOOL using the discharge anomalies obtained in Step 1. The resultant water allocations for each of the 150 AWDUs in the Castile and León Region and every year in the series are reported in Appendix C. The integration of the ensembles of the climate system, water natural system, and water management system further amplifies the database of plausible futures to 258.

Finally, in Step 3, each of these 258 plausible futures and related water allocations to AWDUs are used to force the human system ensemble and produce longitudinal forecasts on the impacts of climate change and water management strategies on income and employment for each of the AWDUs in the DRB. The box-whisker plot in Fig.4 quantifies the uncertainty across the entire basin for each RCP scenario: 2.6, 6.0 and 8.5, for both profit and employment. In Fig 4 (a), the change in profit shows greater dispersion and outliers in the RCP 2.6 scenario, while the RCP 6.0 and RCP 8.5 scenarios display distributions more concentrated around the median. In all cases, the median is negative, indicating a reduction in profits under each scenario. In Fig. 4 (b), changes in employment also exhibit a negative trend in the median across all RCP scenarios. The data dispersion is greater in the RCP 8.5 scenario, followed by RCP 6.0 and RCP 2.6, suggesting increased variability in employment changes

as greenhouse gas concentrations rise. Fig. 4 (a) and (b) reflect an adverse impact on both profit and employment as the RCP scenarios progress, with more pronounced effects in scenarios with higher greenhouse gas concentrations (RCP 8.5).

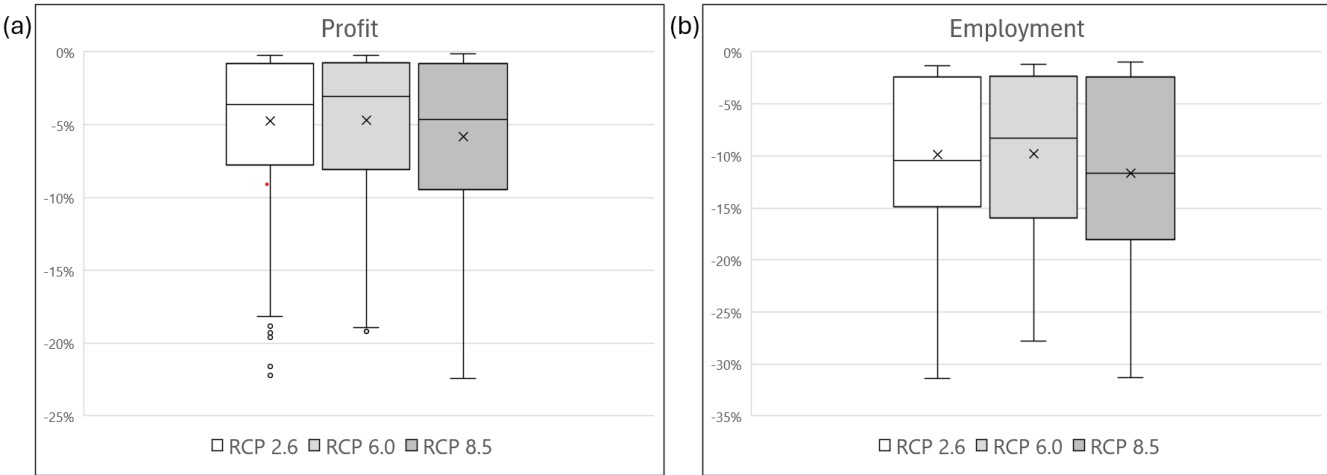

**Figure 4. Box-whisker plots for (a) profit and (b) employment under RCP 2.6, RCP 6.0 and RCP 8.5.**

The box-whisker plot presented lacks spatial disaggregation of the results, thus constraining our understanding of spatial variations within the basin. To address this limitation, Fig. 5 exemplifies the modeling potential in delivering spatially distributed profit outcomes. This figure presents the impacts of climate change on profit for each the five PMP models, considering one adaptive management strategy (M03) and three climatic scenarios (RCP 2.6, 6.0 and 8.5), over the period
2006-2070 and for each of the AWDUs in the DRB. Such spatial representation allows for the identification of regional patterns and trends that remain elusive in the aggregate analysis, which is of value, for local planning and management by pinpointing specific areas necessitating attention or adaptation. Detailed results on the climate change impacts on profit and employment for each AWDU, management strategy, and climatic scenario are available in Appendix D. Additionally, Appendix E provides longitudinal projections of profit and employment for each ensemble model and AWDU.

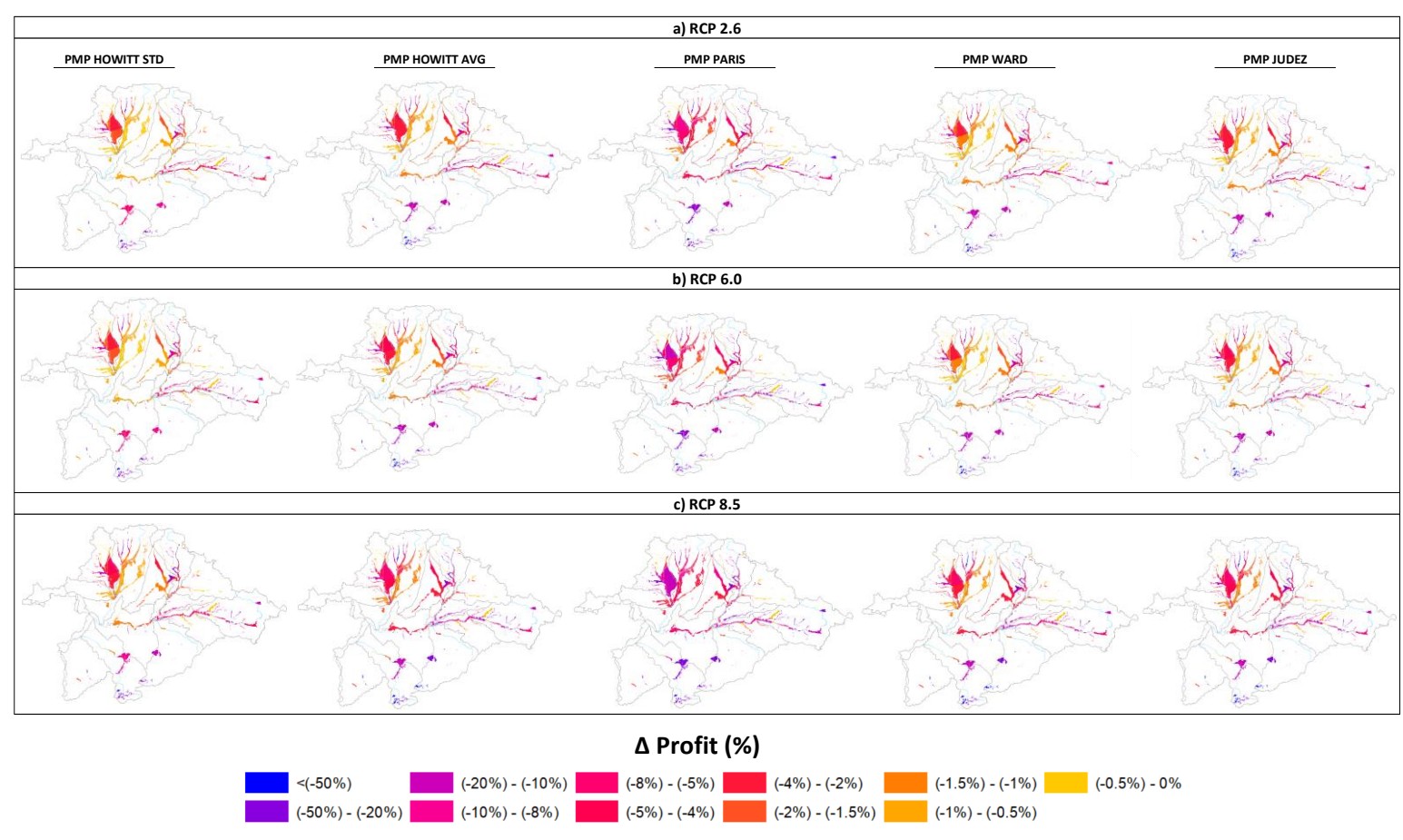

**Δ Profit (%)**

| | | |
|---|---|---|
| <(-50%) | (-20%) - (-10%) | (-8%) - (-5%) | (-4%) - (-2%) | (-1.5%) - (-1%) | (-0.5%) - 0% |
| (-50%) - (-20%) | (-10%) - (-8%) | (-5%) - (-4%) | (-2%) - (-1.5%) | (-1%) - (-0.5%) | |

**Figure 5. Spatially-disaggregated impacts (best estimate) of discharge anomalies under climate change (a) RCP 2.6, b) RCP 6.0, c) RCP 8.5) on profit in the AWDUs of the Douro River Basin for the 2006-2070 periods, considering the management scenario M03. Changes in profit are obtained as the difference between simulated and observed values in year 2017. It is important to note that the GHM MPI-HM yields discharge forecasts that are markedly lower than those obtained with the other models, leading to outliers in the employment and profit predictions, which decrease by nearly 100% in most of the years in the series. This outlier is excluded from the best estimates reported in this figure.**

For all AWDUs, all combinations of models and scenarios predict a reduction in profit and employment due to climate change. Profit and employment losses are higher for 2040-2070, when greenhouse gas concentrations peak and discharge reductions are more marked. Notably, employment and profit losses are significantly higher under the management scenario M01 than in M02, which in turn also displays higher employment and profit losses than M03. This indicates that the expansion of irrigation (incorporated in M01) and its modernization (M01, M02) negatively affect both profit and employment. This is due to the reallocation of water resources from downstream to upstream users, resulting in a cascade of negative and sometimes non linear repercussions that diminish overall profit and employment losses.

AWDUs in the DRB initially manage decreases in water allocations by replacing irrigated crops at the margin (wheat) with rainfed crops, so as to maintain the surface area dedicated to valuable irrigated crops like sugarbeet, vegetables, maize, and fruits. As water allocations continue to decline, AWDUs are constrained to decrease the surface area of increasingly valuable irrigated crops, resulting in more abrupt profit reductions. Due to the labor-intensive nature of these crops, employment also undergoes sudden changes. This explains why marginal decreases in water allocation can lead to substantial, and sometimes disproportionately larger decreases in profit and employment, contributing to a considerable degree of nonlinear change. For instance, AWDUs in the Carrión sub-basin (AWDUs 2000063, 2000064, 2000065, 2000082, 2000083, 2000084, 2000085, 2000086, 2000097, 2000099, 2000105—see Appendix E) can initially manage reductions in water allocation of up to 10% with low to moderate economic losses (<5%). However, they experience abrupt decreases in profit and employment of up to 40% under more stringent water allocation reductions of around 20%. Similarly, AWDUs in the Arlanza sub-basin (AWDUs 2000077, 2000078, 2000079, 2000080, 2000235, 2000320, 2000338, 2000603—see Appendix E), experiencing minimal economic losses (<0.5%) under mild water allocation reductions of 5% or lower, experience abrupt decreases in profit and employment of 12% when water allocation is decreased by 10%. These complex interactions between human behavior and the water system, including nonlinear responses by economic agents to environmental change, cannot be fully understood or modeled without considering the economic system. Coupled socio-ecological modeling is necessary to this end—albeit this can further amplify uncertainty, especially if more than one human system model is used.

## 5 Discussion and conclusions

This paper introduces a modular hierarchy comprising ensembles of socioeconomic and ecological systems (multi-system ensemble). Each ensemble incorporates multiple models (multi-model ensemble) employed to evaluate the repercussions of climate change and management scenarios on water availability, profit, and employment (multi-model ensemble). Using this modeling approach, a comprehensive database of simulations is generated, wherein each result represents the socioeconomic and environmental implications of a distinct combination of scenarios and models, thus quantifying parameter, structural and scenario uncertainties. By integrating human system dynamics into the modeling framework, the resultant grand ensemble

accounts for nonlinearities emerging across both human and water systems, as well as their cascading impacts, thus providing valuable data towards informing robust strategies.

The grand ensemble is built around a commonly used DSS model, AQUATOOL, thus contributing to the generation of actionable science that can be readily adopted by decision makers and other stakeholders. The coupling framework is intentionally crafted to be reproducible and adaptable, with the capability to incorporate alternative climate, hydrologic, DSS, and microeconomic models of farmers that may better represent climate, water, and/or human systems in different regions. Accordingly, our coupling approach and modeling framework can be applied widely and at a relatively low cost by exploiting existing data and/or models. To better inform the replication of our framework elsewhere, we exemplify below what models can be used to populate our framework at each system level.

- Climate system and water natural system: Climate forecasts for alternative climate scenarios are available in climate ensemble experiments, including global (CMIP6, 2023; ISIMIP, 2023) and downscaling (EURO-CORDEX, 2023) experiments at a regional level, which offer simulation outputs for key climate change scenarios such as RCP2.6 or RCP6.0. Managing these simulation outputs requires skills in big data and knowledge of NetCDF and related software, the format in which simulations are reported as well as elemental knowledge about climate model simulation. Climate ensemble experiments can provide relevant data on water discharge, a key input for the water management system. Water discharge data can be alternatively produced using regionally calibrated models, which requires ad-hoc data gathering efforts (albeit some of these models also provide databases for their calibration, as is the case of SWAT (2023)) and modeling skills on the specific software to be used. Also, a list of hydrological models included in Pérez-Blanco (2022) could be incorporated: "the Soil and Water Assessment Tool – SWAT (Arnold et al., 1998); Annualized Agricultural Non-Point Source Pollution Model – AnnAGNPS (Young et al., 1989); Areal Nonpoint Source Watershed Environment Response Simulation – ANSWERS 2000 (Bouraoui and Dillaha, 2000); Agricultural Policy / Environmental eXtender Model – APEX (Gassman et al., 2009); US Army Corps of Engineers - Hydrologic Engineering Center - Hydrologic Modeling System – HEC-HMS (US Army Corps of Engineers, 2015); and Soil and Water Integrated Model – SWIM (Krysanova et al., 2005)".

- Water management system: The data inputs necessary to run the relevant DSS in a given basin are typically accessible to the competent authority, either directly or through a consulting company. Some widely used DSS that could be incorporated into our modeling framework include AQUATOOL, WEAP, TOPKAPI, MIKE, RIBASIM or LISFLOOD. Critically, DSS are often profusely edited to account for the unique features of the basin at hand, and thus their management requires support from an expert. In our illustrative example with AQUATOOL, the competent authority was the DRB, which typically relies on an external consultant to run hydrologic simulations on the

management and allocation of water resources. For this research, USAL collaborated with the consultant to develop the coupling and run the simulations, leveraging on funding provided by the DRB.

-Human system: The human system can be populated by any mathematical programming model of agricultural water use available in the literature. The data necessary to run these models is provided in Appendix B. In the case of EU river basins, all necessary data is publicly available although granularity may differ. For example, in the case of the Portuguese part of the DRB, a similar database to the one used in this paper is publicly available albeit the granularity is significantly lower (regional rather than AWDU scale).

The current and previous (Gil-García et al., 2023; Pérez-Blanco et al., 2021a, 2021b) versions of the AQUATOOL-based human-water system DSS presented in this paper have been already used by stakeholders in the context of financial and economic viability assessments of new water works proposed in the Douro River Basin Plan under climate change and uncertainty, including La Rial Dam, Los Morales Dam, or the Lastras de Cuéllar Dam (assessed with previous versions of the model with a focus on input uncertainty), as well as the Las Cuezas dams (assessed with the current version of the model that includes structural uncertainties in models). All of these assessments were commissioned by the river basin authority.

Future scientific research offers several avenues to further develop and expand the proposed hierarchical coupling framework for understanding the intricate interactions between human actions and water resources, and related uncertainties. Firstly, it is possible to introduce improvements to the individual models within each module by integrating recent scientific advancements within each discipline. For instance, recent developments in microeconomic modeling decouple land use choices from water use choices, allowing for two rather than one decision variables as it is usually the case in conventional PMP and other mathematical programming models, where agents only decide on land use (Graveline and Merel, 2014; Loch et al., 2020; Sapino et al., 2022a). This allows for the representation and assessment of adaptation measures at the intensive margin (such as deficit/supplementary irrigation), going beyond the extensive (transition to less water-intensive crops) and super-extensive (transition to rainfed crops) adaptations examined in conventional PMP and other mathematical programming models.

Secondly, additional protocols could be introduced across systems to bolster the framework and its interactions. For instance, this might entail incorporating distinct protocols for water use decisions independent of land use decisions in the coupling between the human and hydrologic modules.

Thirdly, the modeling framework could benefit from the integration of extra modules, including the linkage with macroeconomic or crop models. This integration would allow for the evaluation of climate change impacts on crop yields (using crop models) and prices (using macroeconomic models). Multi-model ensembles of Global Gridded Crop Models are available in ISIMIP (2023) and could be coupled following a similar procedure as the one described here for the water natural system/GHMs, while the integration of macroeconomic models such as Computable General Equilibrium (CGE) models

(Hertel and Liu, 2016) into water systems research has been already (Parrado et al., 2020; Pérez-Blanco et al., 2021a; Pérez-Blanco and Gutiérrez-Martín, 2017; Ronneberger et al., 2009). By adding these and other new system modules, uncertainty would further cascade across systems. This can help us in identifying new vulnerabilities and further underpin robust decision making.

On the other hand, as the number of modules, protocols, structures, and other modeling factors (inputs, parameters, structures) considered in the analysis grow, other issue other issues may arise that may reduce the tractability of the problem. These issues are dealt with in our last three recommendations for improvement.

   Fourthly, having incorporated multiple uncertainties, the model output may vary "so wildly as to be of no practical use" (Saltelli et al., 2008). However, as noted by Saltelli et al. (2008) and in line with previous work by Beven and Binley (1992) and Beven
and Freer (2001) that introduced the equifinality concept (i.e., distinct configurations of model components such as inputs, parameters, or structures, can lead to similar or equally acceptable representations of the real-world process of interest), this "trade-off may not be as dramatic as one might expect, and that increasing the number of input factors does not necessarily lead to an increased variance in model output." Typically, a few inputs create almost all the uncertainty, and the majority make a marginal contribution.

Fifthly, computational costs may pose a challenge in conducting uncertainty quantification and analyses where 1) each model run demands a considerable amount of time, stretching from minutes to hours or even longer, especially in the case of highly intricate models; and/or 2) where the model encompasses numerous uncertain inputs, which expand exponentially the computational cost with the increase in the number of inputs—a phenomenon known as the curse of dimensionality. Addressing computational expenses is crucial in many practical sensitivity analyses and model intercomparison projects.
Strategies to mitigate this burden include employing emulators or metamodels driven by machine-learning techniques that are particularly suitable for large models (Storlie et al., 2009), and employing screening methods to reduce the dimensionality of the problem such as High-dimensional model representations (Li et al., 2006).

   Sixthly, at some point, modelers must decide on the boundaries for the uncertainty quantification, i.e., the inputs, models, etc. and that will condition the outputs of the modeling exercise. This involves defining some limits to not generate computational
costs we cannot afford through model selection and other techniques. For example, techniques for model selection can be employed to assign weights to the models in the ensemble based on their performance in calibration and forecasting errors. This could help us not only reduce computational costs (e.g., by discarding some models) but also reduce potential biases, such as the simulation outputs from GHM MPI-HM, which yields discharge forecasts that are markedly lower than the other models (with reductions close or equal to 100%), shifting the ensemble spread downwards and with important implications for
human system forecasting (see Fig. 5). It should be noted that while model selection techniques based on forecasting errors can be implemented for GHM and GCM, the measurement of forecasting errors in human system models is significantly more

challenging: information on agents' crop choices in the DRB is available only since 2004, which leads to a significantly more reduced data series that complicates the implementation of rolling origin or other techniques to measure forecasting errors. On the other hand, the adoption of model selection techniques on the basis of calibration errors can be misleading, since models with higher calibration errors may show better predictive performance (Pindyck, 2015).

The convenience of adopting model selection techniques is a question for debate, since weighting can significantly affect modeling results and condition stakeholder choices and decision making. On the other hand, it has been contended that whenever model selection techniques are not considered, each potential simulation outcome is equally important, "which can also be interpreted as an implicitly equal weighting" (Taner et al., 2019). This is more so when results are explicitly reported using best estimates as done in Fig. 3-5. While this statistical treatment is a key step in making results understandable to users, it may introduce nontrivial biases through the processing and communication of modeling results that has to be explicitly addressed by the use of dispersion measures (Fig. 5) and with the development of adequate processing techniques for modeling outputs (e.g., serious games that convey the economic and environmental repercussions of water extremes). On the other hand, a similar critique can be made to weighting. The decision of whether to assign weights to simulation outcomes or leave them open for interpretation remains a subject of debate among academics (Taner et al., 2019).

**Appendix A: New infrastructures in the Douro River Basin Plan**

| AWDU | System | Situation | Scenario 1 | Scenario 2 | Scenario 3 |
|---|---|---|---|---|---|
| 2000003 | Esla | Irrigation modernization | × | × | |
| 2000006 | Esla | Irrigation modernization | × | × | |
| 2000014 | Órbigo | Irrigation modernization | × | × | |
| | | La Rial and Los morales reservoirs | × | × | × |
| 2000017 | Órbigo | Irrigation modernization | × | × | |
| | | La Rial and Los morales reservoirs | × | × | × |
| 2000018 | Órbigo | Irrigation modernization | × | × | |
| | | La Rial and Los morales reservoirs | × | × | × |
| 2000023 | Órbigo | Irrigation modernization | × | × | |
| | | La Rial and Los morales reservoirs | × | × | × |
| 2000025 | Tera | Irrigation modernization | × | × | |
| 2000026 | Tera | Decrease in irrigated areas | × | × | × |
| 2000034 | Esla | Expansion of irrigated areas | × | | |
| 2000038 | Órbigo | Irrigation modernization | × | × | |

| ID | Basin | Measure | | | |
|----|-------|---------|---|---|---|
| | | La Rial and Los morales reservoirs | × | × | × |
| 2000041 | Esla | New AWDU in 2027 | × | × | × |
| | | Balsa Sector IV reservoir to AWDU 2000033 | × | × | × |
| 2000047 | Esla | Expansion of irrigated areas | × | | |
| 2000049 | Tera | New AWDU in 2027 | × | × | × |
| | | Expansion of irrigated areas to AWDU 2000025 | × | | |
| 2000052 | Órbigo | Irrigation modernization | × | × | |
| | | La Rial and Los morales reservoirs | × | × | × |
| 2000054 | Esla | New AWDU in 2033 | × | × | × |
| | | Valcuende de Almanza reservoir to AWDU 2000040 | × | × | × |
| 2000055 | Esla | Expansion of irrigated areas | × | | |
| | | Vallehondo reservoir | × | × | × |
| 2000057 | Esla | Expansion of irrigated areas | × | | |
| 2000064 | Carrión | Irrigation modernization | × | × | |
| | | La Cueza 1 and La Cueza 2 reservoirs | × | × | × |
| 2000065 | Carrión | Irrigation modernization | × | × | |
| | | La Cueza 1 and La Cueza 2 reservoirs | × | × | × |
| 2000071 | Pisuerga | Expansion of irrigated areas | × | | |
| | | Burejo reservoir | × | × | × |
| 2000073 | Pisuerga | Expansion of irrigated areas | × | | |
| | | Las Cuevas reservoir | × | × | × |
| 2000080 | Arlanza | Expansion of irrigated areas | × | | |
| | | Irrigation modernization | × | × | |
| | | Castrovido reservoir | × | × | × |
| 2000082 | Carrión | Irrigation modernization | × | × | |
| | | La Cueza 1 and La Cueza 2 reservoirs | × | × | × |
| 2000083 | Carrión | Irrigation modernization | × | × | |
| | | La Cueza 1 and La Cueza 2 reservoirs | × | × | × |
| 2000091 | Bajo Duero | Irrigation modernization | × | × | |
| 2000092 | Bajo Duero | Irrigation modernization | × | × | |
| 2000094 | Bajo Duero | Irrigation modernization | × | × | |
| 2000097 | Carrión | Expansion of irrigated areas | × | | |

| ID | Basin | Description | | | |
|---|---|---|---|---|---|
| | | La Cueza 1 and La Cueza 2 reservoirs | × | × | × |
| 2000100 | Pisuerga | Expansion of irrigated | × | | |
| | | Boedo reservoir | × | × | × |
| 2000102 | Pisuerga | Valles de Cerrato reservoir | × | × | × |
| 2000108 | Bajo Duero | Irrigation modernization | × | × | |
| 2000122 | Alto Duero | Irrigation modernization | × | × | |
| 2000128 | Alto Duero | Expansion of irrigated areas | × | | |
| | | Irrigation modernization | × | × | |
| 2000132 | Alto Duero | Expansion of irrigated areas | × | | |
| | | Dor reservoir | × | × | × |
| 2000143 | Alto Duero | Expansion of irrigated areas | × | | |
| 2000166 | Cega-Eresma-Adaja | New AWDU in 2033 | × | × | × |
| | | Torreiglesias reservoir to AWDU 2000159 | × | × | × |
| 2000168 | Cega-Eresma-Adaja | Expansion of irrigated areas | × | | |
| | | Lastras de Cuéllar reservoir | × | × | × |
| 2000171 | Cega-Eresma-Adaja | New AWDU in 2033 | × | × | × |
| | | Carbonero, Cigueñuela, Lastras de Cuéllar and Torreiglesias reservoirs to AWDU 2000168 and 2000164 | × | × | × |
| 2000202 | Águeda | Irrigation modernization | × | × | |
| 2000207 | Tormes | New AWDU in 2027 | × | × | × |
| | | Expansion of irrigated areas to AWDU 2000208 | × | | |
| 2000209 | Tormes | Expansion of irrigated areas | × | | |
| | | Gamo reservoir | × | × | × |
| 2000210 | Tormes | Expansion of irrigated areas | × | | |
| | | Margañán reservoir | × | × | × |
| 2000211 | Tormes | Decrease in irrigated areas | × | × | × |
| 2000212 | Águeda | New AWDU in 2027 | × | × | × |
| | | Expansion of irrigated areas to AWDU 2000185 | × | | |
| 2000213 | Esla | New AWDU in 2027 | × | × | × |
| | | Expansion of irrigated areas to AWDU 2000202 | × | | |

| 2000280 | Órbigo | Expansion of irrigated areas | × | | |
| | | Irrigation modernization | × | × | |
| 2000282 | Esla | New AWDU in 2033 | × | × | × |
| | | Balsa Sector V reservoir to 2000033 | × | × | × |
| 2000598 | Órbigo | Irrigation modernization | × | × | |
| | | La Rial and Los morales reservoirs | × | × | × |
| 2000600 | Órbigo | Irrigation modernization | × | × | |
| | | La Rial and Los morales reservoirs | × | × | × |
| 2000605 | Cega-Eresma-Adaja | New AWDU in 2033 | × | × | × |
| | | Carbonero, Cigueñuela, Lastras de Cuéllar and Torreiglesias reservoirs to AWDU 2000164 | × | × | × |
| 2000606 | Cega-Eresma-Adaja | New AWDU in 2033 | × | × | × |
| | | Torreiglesias reservoir to AWDU 2000159 | × | × | × |
| 2000607 | Cega-Eresma-Adaja | New AWDU in 2033 | × | × | × |
| | | Lastras de Cuéllar reservoir to AWDU 2000168 | × | × | × |
| 2000608 | Cega-Eresma-Adaja | New AWDU in 2033 | × | × | × |
| | | Carbonero and Ciguiñuela reservoirs to AWDU 2000164 | × | × | × |

**Appendix B: Data input of the PMP models**

*[see Excel file attached to the submission for the database]*

**Appendix C: Data of water allocations for each AWDUs**

*[see Excel file attached to the submission, 2 sheets – 1st monthly allocation, 2nd yearly water allocation deficit (input for micro model)]*

 **Appendix D: Spatial distribution of profit and employment for each ensemble model**

(a)

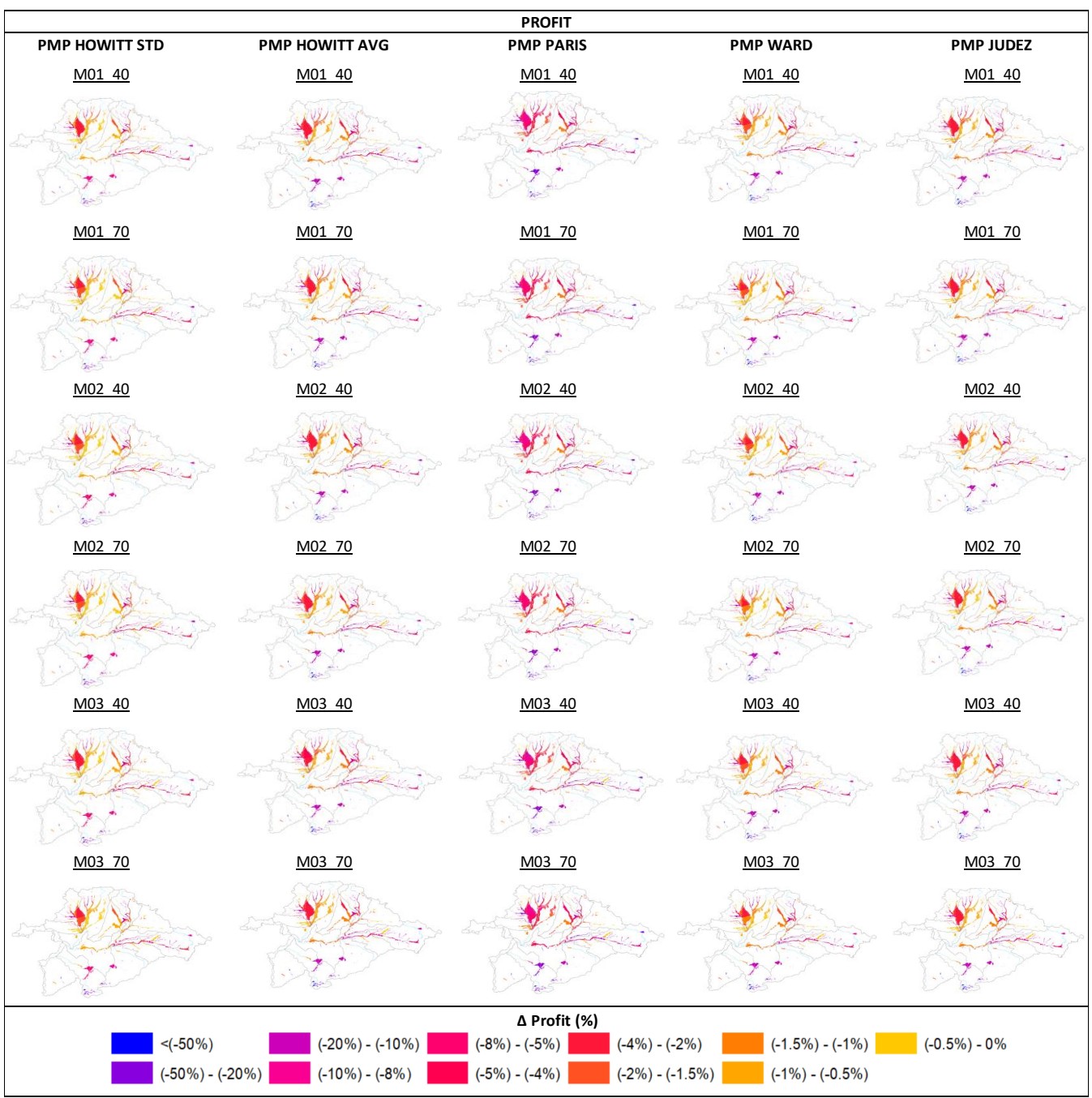

(b)

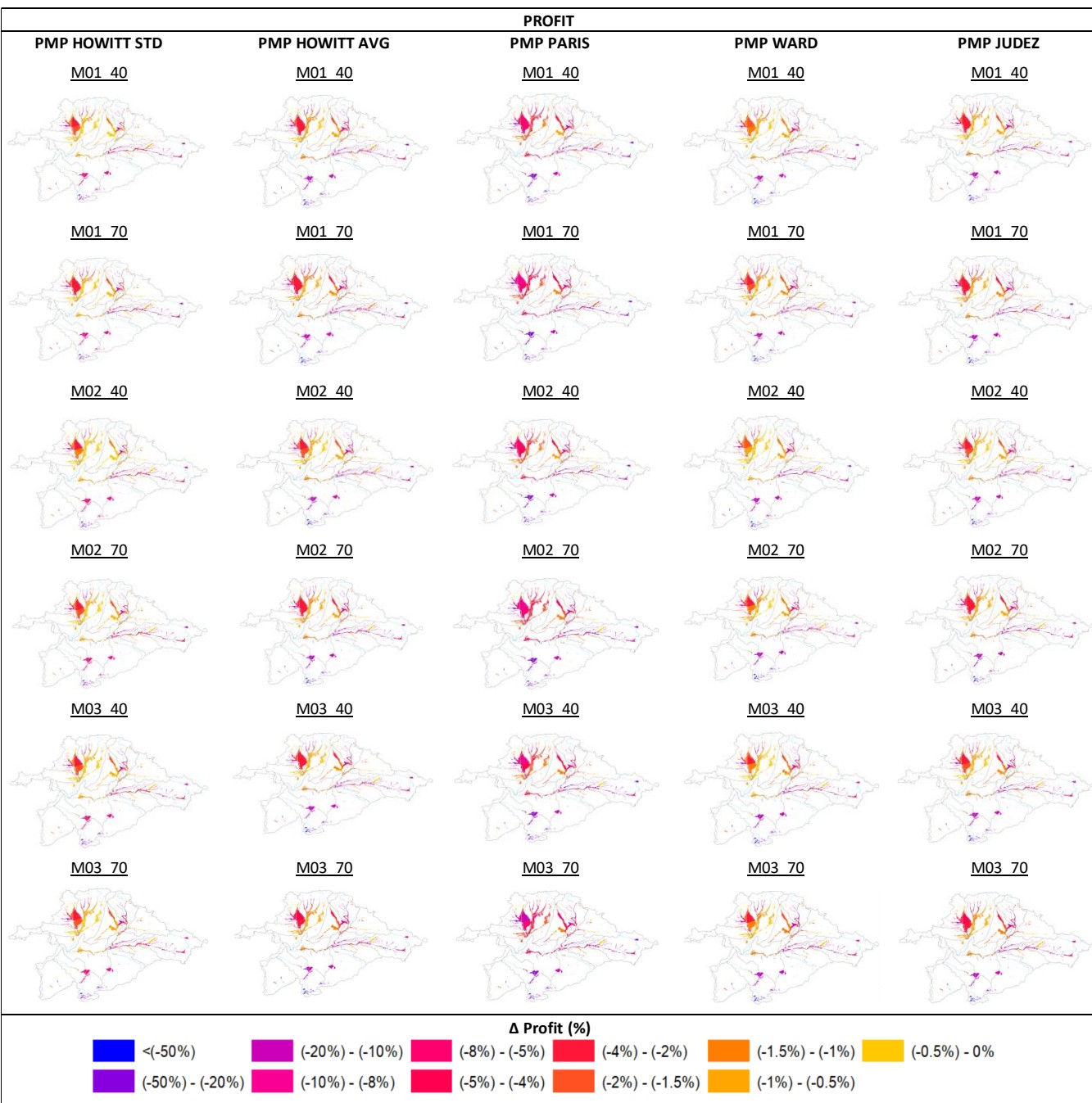

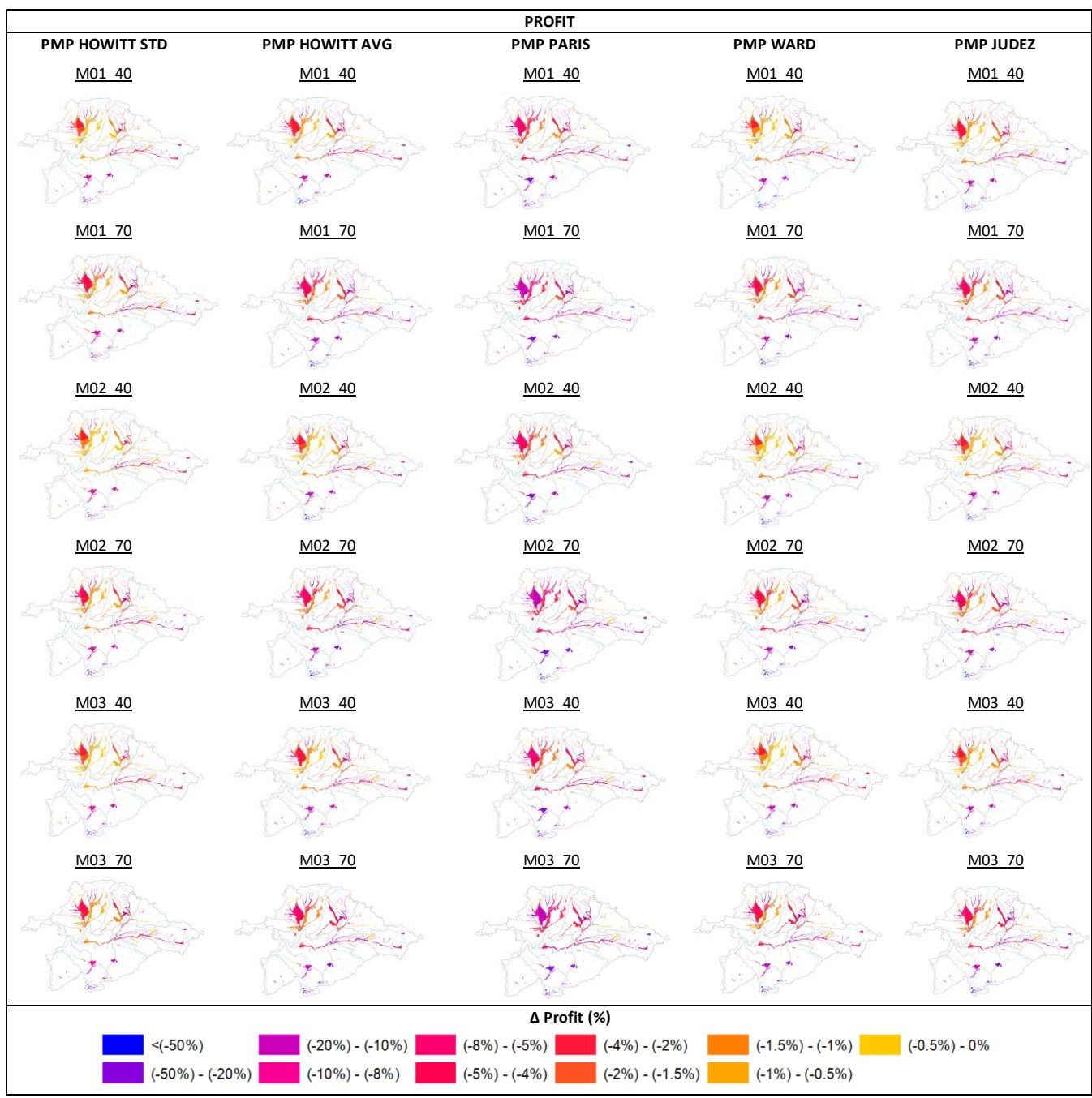

**Figure 1. Spatially-disaggregated impacts (best estimate) of discharge anomalies under climate change (a) RCP 2.6, b) RCP 6.0, c) RCP 8.5) on profit in the AWDUs of the Douro River Basin for the 2006-2040 and 2040-2070 periods. Changes in profit are obtained as the difference between simulated values under alternative climate change and management scenarios and observed values in year**

**2017. It is important to note that the GHM MPI-HM yields discharge forecasts that are markedly lower than those obtained with the other models, leading to outliers in the employment and profit predictions, which decrease by nearly 100% in most of the years in the series. This outlier is excluded from the best estimate reported in this figure.**

(a)

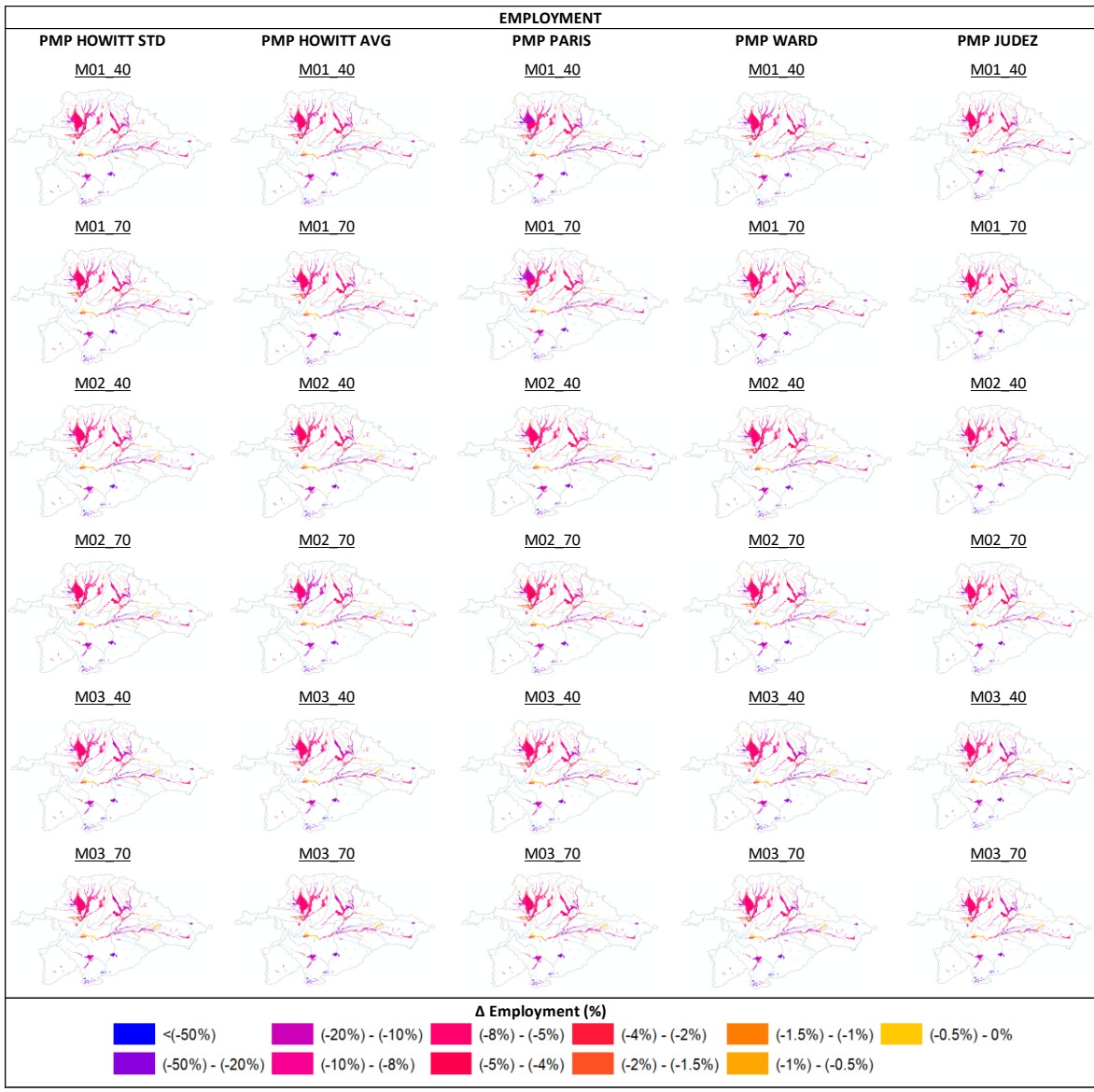

(b)

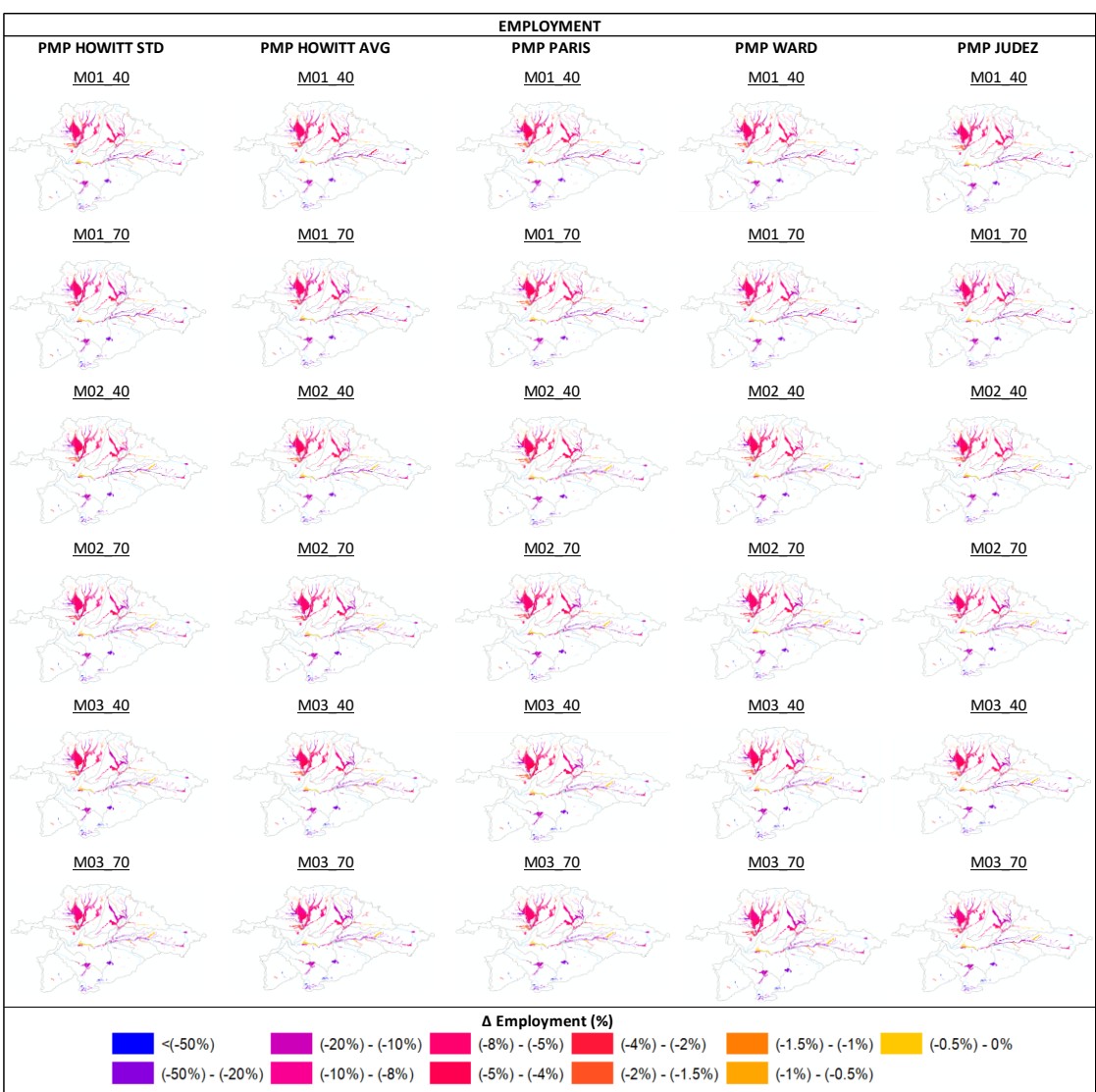

(c)

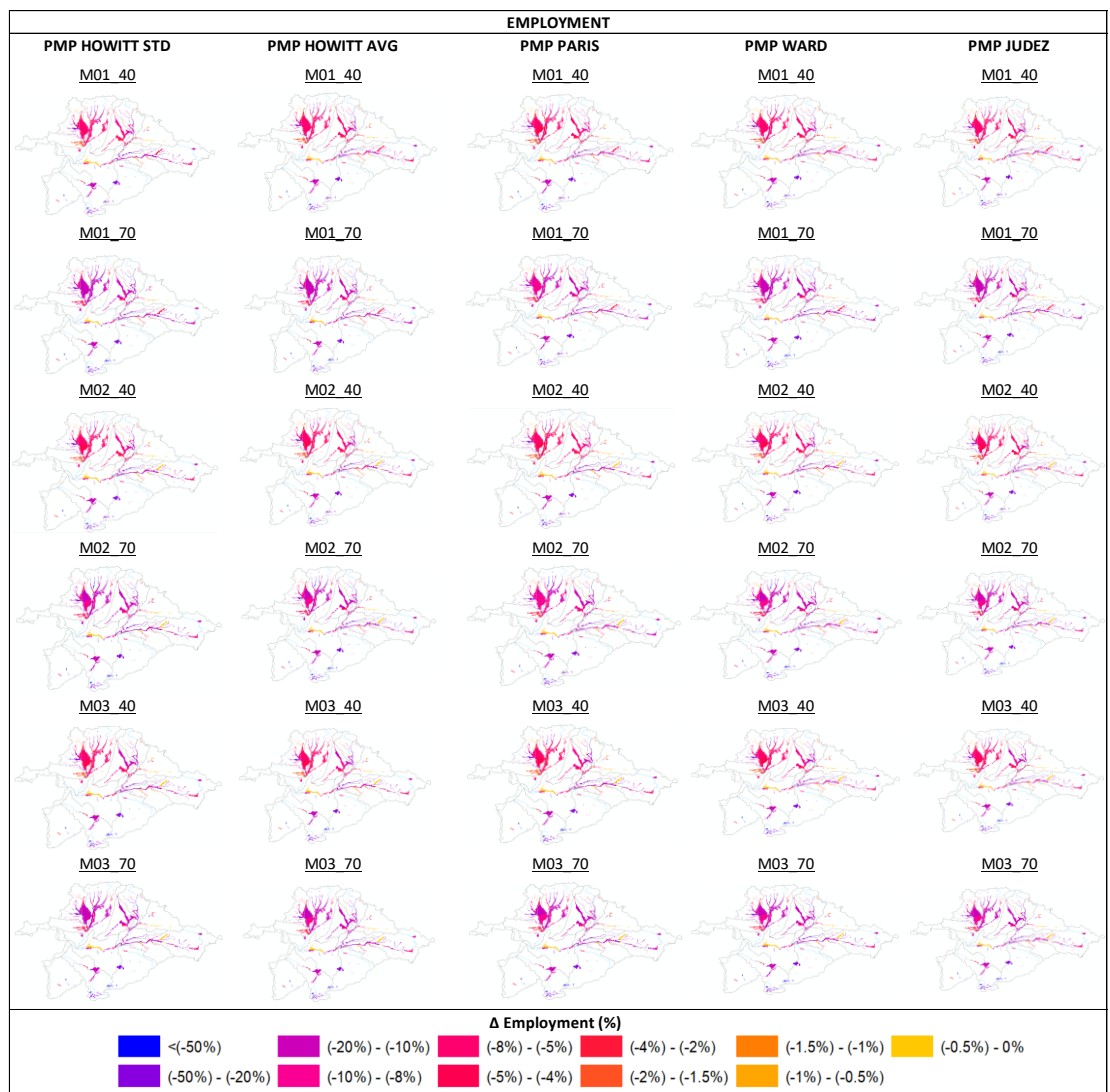

**Figure 2. Spatially-disaggregated impacts (best estimate) of discharge anomalies under climate change (a) RCP 2.6, b) RCP 6.0, c) RCP 8.5) on employment in the AWDUs of the Douro River Basin for the 2006-2040 and 2040-2070 periods. Changes in employment are obtained as the difference between simulated values under alternative climate change and management scenarios and observed values in year 2017. It is important to note that the GHM MPI-HM yields discharge forecasts that are markedly lower than those obtained with the other models, leading to outliers in the employment and profit predictions, which decrease by nearly 100% in most of the years in the series. This outlier is excluded from the best estimates reported in this figure.**


**Appendix E: Data of the impacts on profit and employment for the AWDUs**

*[see Excel file attached to the submission]*

**Author contribution**

**Laura Gil-García**: Data Curation, Formal analysis, Software, Investigation, Visualization, Writing – Original Draft, Writing – review & editing; **Nazaret M. Montilla-López:** Data Curation, Software, Investigation, Writing- review & editing; **Carlos Gutiérrez-Martín**: Supervision, Software, Investigation, Writing – review & editing; **Ángel Sánchez-Daniel:** Data Curation,
Software; **Pablo Saiz-Santiago:** Data Curation, Software; **Josué M. Polanco-Martínez:** Data curation, Writing – review & editing; **Julio Pindado:** Writing – review & editing; **C. Dionisio Pérez-Blanco:** Conceptualization, Funding acquisition, Project administration, Supervision, Investigation, Methodology, Resources, Writing – Original Draft, Writing – review & editing.

**Competing interests**

The authors declare that they have no conflict of interest.

**Acknowledgements**

This research is part of the project TED2021-131066B-I00, funded by MCIN/AEI/10.13039/501100011033 and by the European Union "NextGenerationEU"/PRTR; and of the PRIMA Foundation's TALANOA-WATER Project (Talanoa Water Dialogue for Transformational Adaptation to Water Scarcity under Climate Change).

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
