# Peer review of "Actionable human-water systems modeling under uncertainty"

_Hydrology and Earth System Sciences, 2024_

## Author Response (AR1)

**Comments from referee 1**

This is a good paper that is well structured, argued and delivered. The combined models remain a contribution in the space and the justification for its development and design are well supported. The insights from both the study and its applicability to a range of contexts, decision-makers, and stakeholders—clearly articulated in the replication text— is highly useful and a little unusual in economics. Liked that a lot.

My only real concern then is the use of the term uncertainty here. If we assume a Knightian approach to the ideas, as I would usually so that we're all clear on my stance here, then uncertainty is the consummate unknown in that we are not even aware we are unaware. If, as stated here, the concept of probabilities and data can be used to construct scenarios and outcomes that parameterise the conditions then we are dealing with risk. The distinction is important when building these ideas out and analysing them in such a manner as detailed here. And they make a big difference to the interpretation and usefulness of the results. In my view you can't have a DSS built on uncertainty because it is unknown and as such cannot be parameterised-the whole point of the term.

Thus, I would like the authors to explain clearly why they are comfortable with this approach or if they agree with my views and will update the term use and constructs. It would want to be a very good argument if they are to convince me the existing approach is appropriate. As an author in this area, copping a lot of flak from engineers on this exact topic, the argument should be made, agreed with, and then worked back into a subsequent version of the paper.

Otherwise, the work is solid and well-constructed/nicely written up. I congratulate the authors and wish them well with the paper going forward.

Moderate revisions needed.

**Author's response to referee 1**

*We thank you for your time and dedication to our paper and your detailed review. We agree that the distinction among types of uncertainty, and between uncertainty and risk, is crucial in modeling and decision analysis, and we appreciate you highlighting this point, because it allows us to introduce some clarifications that we agree were necessary.*

*Multiple definitions of uncertainty have been used, including that of Knight (1921), where uncertainty is defined as "a lack of any quantifiable knowledge about some possible occurrence, as opposed to the presence of quantifiable risk". Since Knightian uncertainty is not quantifiable, may not be adequate–as you duly note. On the other hand we have probabilistic risk, where we know what plausible futures lie ahead of us as well as the associated probabilities. There are also some shades of uncertainty that lie in between probabilistic risk and Knightian uncertainty. For example, it may be possible that we know some scenarios may happen in the future, but we do not know their probabilities. This is more limiting than probabilistic risk but less limiting than Knightian uncertainty, and in principle could be modeled (Walker et al., 2003). A good example of this are climate simulations by the IPCC/CMIP6, where RCP/SSP scenarios are simulated without knowing their probability of occurrence.*

*In this paper we adopt the definition of uncertainty provided by Walker et al. (2003), who identify different levels across the uncertainty spectrum: 1) determinism (where point*

*predictions are reliable), 2) probabilistic risk, 3) (deep) uncertainty type 1 (we do not know what inputs, parameters and/or model structures are right, nor their probability, but we can anticipate how the system will react to these), 4) (deep) uncertainty type 2 (we know we do not know), and 5) complete ignorance (we are not aware of what we do not know). Knightian uncertainty would fall in the levels 4-5, which precludes modeling. But deep uncertainty type 1 can be modeled. This is the type of uncertainty that models typically address when dealing with uncertainty. Examples of this are the model intercomparison projects ISIMIP/CMIP/LUMIP/AGMIP/HEPEX.*

*Walker et al. (2003) define uncertainty as a situation where "1) it is not possible to identify all plausible futures, or 2) assign a probability to each identified plausible future". Point 2) refers to deep uncertainty type 1, which can be modeled, while point 1) cannot (Knightian uncertainty). Note that this definition explicitly excludes probabilistic risk. In our study, as other model intercomparison experiments do, we focus on deep uncertainty point 1. Admittedly, our modeling does not offer probabilities of occurrence, but it nonetheless provides a valuable tool for decision-makers. We go as far as we can go with the models and information we have, and offer this to decision makers so that they can design policies leveraging the best possible modeling outputs available (including uncertainty) and their own expert judgement.*

*We appreciate your feedback and have carefully considered and addressed your comments when revising and improving our work in the new version of the article. We have added a definition of uncertainty the first time the word is mentioned. We have moreover added a footnote clarifying key uncertainty concepts.*

*Walker, W.E., Harremoës, P., Rotmans, J., Van Der Sluijs, J.P., Van Asselt, M.B.A., Janssen, P., Krayer Von Krauss, M.P., 2003. Defining Uncertainty: A Conceptual Basis for Uncertainty Management in Model-Based Decision Support. Integrated Assessment 4, 5–17. https://doi.org/10.1076/iaij.4.1.5.16466*

**Comments from referee 2:**

I agree with the underlying premise of this paper – that the use of DSS's by policymakers needs to include an extensive scenario analysis to explore the uncertainty (or confidence) in the outputs. The authors appear to claim that they are doing a much better job of encompassing all uncertainties than has been done previously. I would question this. There has been a lot of work done on uncertainty and scenario analysis. The authors may be able to claim that theirs is the best approach so far, but this is merely claimed – there is no evidence to support this. While obtaining such evidence is intrinsically impossible, a more comprehensive literature review that discusses the various approaches that have been employed so far (e.g. Bayesian networks, coupled complex models, agent-based models) may help give credibility to the authors' claims. While not necessarily relevant, the author may find the following papers useful:

- Hamilton et al (2019) A framework for characterising and evaluating the effectiveness of environmental modelling, Environmental Modelling and Software 118, 83-98 https://doi.org/10.1016/j.envsoft.2019.04.008

- Maier et al. (2016) An uncertain future, deep uncertainty, scenarios, robustness and adaptation: How do they fit together? Environmental Modelling and Software, 81, 154-164. https://doi.org/10.1016/j.envsoft.2016.03.014
- Guillaume, J., "Designing a knowledge system for managing deep uncertainty?" (2022). International Congress on Environmental Modelling and Software. 12. https://scholarsarchive.byu.edu/iemssconference/2022/Stream-D/12

These are just papers that I am familiar with (for the record, I know the authors, but I am not a co-author of these papers).

One question is whether the authors have just created another DSS that includes assessment of uncertainty or if this is actively being used by policymakers. Is there any evidence that the DSS is actually being used? If not, this is just another study in the academic arena and doesn't address the lack of uptake by policymakers. It is reassuring that there is an author who is not an academic on this paper, but there have been other papers that include non-academic authors and this by itself doesn't necessarily result in the adoption of the work by policymakers.

I think the paper should be framed as an example of how to improve DSSs by taking more careful consideration of uncertainty, including consideration of multiple scenarios.

**Author's response to referee 2**

*We appreciate your comments and suggestions, which provide relevant insights we have incorporated into our work. We have read the articles you mentioned, as well as many others, to better contextualize our research as per your first comment. In fact, before and during the review at HESS, some of the authors of the present study were engaged in a review of uncertainty analysis in hydroeconomic models that was recently concluded, and has given us timely insights into current practices for uncertainty quantification in human-water systems modeling.*

*https://transcend.usal.es/deliverable-3-1-white-paper-methods-models-for-assessing-policy-performance-under-deep-uncertainty/*

*First, let us clearly state we do not believe our work is "much better" than previous research. Our work complements and builds on valuable previous research, some of which has made unparalleled step changes in the literature that have fundamentally transformed our view of human-water systems and uncertainty quantification / analysis. Also please note our paper is primarily an applied paper that aims at addressing some of the key gaps in uncertainty quantification in human-water systems identified in the literature, including in the papers you mention by Hamilton et al (2019), Maier et al. (2016) and Guillaume (2022), but also by Saltelli (2019) and Puy (2022), among others.*

*We focus on a specific gap in the literature, namely the limited quantification of structural uncertainties in human-water system models, and develop an approach that we believe can contribute to address it. In the recent overview of hydroeconomic models we mentioned before, it was found that out of 198 papers in the sample only 7 quantified structural uncertainties (as compared to 148 quantifying input uncertainty and 40 parmeter uncertainty). Of these studies, 51 included a DSS or water resources management model such as WEAP, of which only 3 quantified structural uncertainties (and partially, i.e., in only one of the systems). Moreover, not a single paper quantified uncertainties in both human and water systems (i.e., studies quantified uncertainties*

*either in human or water systems). While examples of multi-model/model intercomparison experiments to quantify structural uncertainties exist in the hydrological literature (e.g., HEPEX), their application in coupled human-water systems is limited. This is also observed in the wider natural resources literature, where multi-system model-intercomparison experiments to quantify structural uncertainties address only ecological and not human systems (CMIP, ISIMIP, AGMIP, etc.). This gap is largely attributable to human systems modeling: while model intercomparison/multi-model experiments have become a fundamental tool to quantify structural uncertainty in ecological (including water) systems research, they are rare in human systems or SES research.*

*To address this gap, in our study we propose a multi-system model intercomparison experiment across climate, human and water systems.*

*The paper has been improved to acknowledge the relevant work made by others and better place our contribution in this context. To this end, we now briefly present the existing literature on uncertainty quantification in human-water systems, identify the main gaps with a focus on structural uncertainties, and cite key papers. For more detailed information, we direct the reader to a recent review that systematically reviews uncertainty quantification in human-water systems. We also critically address in the discussion the limitations of our work, the most relevant being that it focuses on structural uncertainties with a partial assessment of input uncertainty (via climate change scenarios) and without addressing parameter uncertainties. We also discuss the challenge of combining model intercomparison projects to quantify uncertainties with global sensitivity analyses typically used to quantify input and parameter uncertainties, due to the significant computational cost.*

*The DSS upon which the present study builds its ensemble approach, AQUATOOL, is the software used by Spanish river basins to plan and manage watersheds, specifically in the Duero River Basin. The current and previous versions of the human-water system model built around AQUATOOL and presented in this paper has been also used by stakeholders, albeit admittedly for specific purposes related to financial and economic viability assessments of new water works proposed in the plan, and not for day to day river basin planning. Examples of applications of the proposed model include the economic and financial feasibility assessment of the La Rial Dam, Los Morales Dam, or the Lastras de Cuéllar Dam (assessed with previous versions of the model)* (Gil-García et al., 2023; Pérez-Blanco et al., 2021a, 2021b), *as well as the Las Cuezas dams (assessed with the current version of the model that includes structural uncertainties in models), all of which were commissioned by the river basin authority. The model presented in this paper has been also used to inform the co-design of transformational adaptation policies with stakeholders, including river basin authorities, in the context of the TALANOA-WATER project ([https://talanoawater.com/](https://talanoawater.com/)). We now mention all these applications that illustrate the potential of the model in Section 5.*

*Below we address your specific comments, one by one.*

Specific comments

1. Page 1, line 14: Are you sure that you thoroughly quantified and assessed the uncertainty? Is there no possibility that you missed a source of uncertainty? Recommend deleting "thoroughly" as it is not really needed.

*Thank you. We have deleted "thoroughly" as suggested. This was indeed*
*inaccurate since we are focusing on a specific source of modelling uncertainties,*
*namely structural uncertainty (and only partially on input uncertainty).*

2. Page 2: Font is far too small in some of the panels. Suggest simplifying the panels
and increasing the font size.

*We have expanded the font size of the graphic.*

3. Page 2, line 32: I would delete "(nonlinear change)" as it is not really necessary
in this sentence. Also "unexpected, sometimes abrupt, change" would be better.

*We have replaced "(nonlinear change)" by "unexpected, sometimes abrupt,*
*change" as suggested.*

4. Page 2, lines 35-36: I think "that gives a false appearance of uncertainty
reduction" would be better phrasing.

*We have replaced "that artificially reduces uncertainty" by "that gives a false*
*appearance of uncertainty reduction" as suggested.*

5. Page 3, line 46: "Parameters" would be better than "constants" here as calibrating
constants would mean they are not constant. This would also agree with use of
"parameter" later in the paper (e.g. lines 49, 51)

*We have replaced "constants" by "parameters" as suggested.*

6. Page 3, lines 56-57: Suggest stating the papers cited here are examples.

*This has been amended as suggested.*

7. Page 3, lines 68-69: I would question this in terms of DSS. In terms of
policymakers and what they use for planning and management, then maybe, but
DSS themselves have been explored using ensemble research. I agree with the
statement in the following sentence, but this sentence misses the mark. I suggest
deleting it.

*We have deleted this sentence.*

8. Page 3, line 74: not "concealed" as this implies that academics are hiding these
methods. "confined" would be better.

*We have replaced "concealed" by "confined" as suggested.*

9. Page 4, lines 97-100: are these numbers known to 6 or 7 significant figures? I
would think the uncertainty in these values would be considerably larger than 0.1
million m3/year.

*Thank you for the comment. This is information published by the River Basin Authority in its basin plan, but we agree it seems overconfident. We have removed the decimal numbers to reflect this consideration.*

10. Page 4, lines 99-100: Having these periods overlap (1940-2005 and 1980-2005) is not ideal. Better to give the resources from 1940-1979 and 1980-2005.

*The periods 1940-2005 and 1980-2005 are standard periods used by the Spanish basin authorities (including the Duero River Basin Authority) in their river basin plans and special drought plans to calculate the basin's average annual supply (DRBA, 2018), as well as to run simulations using AQUATOOL. The period 1940-2005 is identified as the "long series" and the period 1980-2005 as the "short series". Although we agree with you, we decided to stick to these standards to enhance the actionability of the model.*

11. Page 4, line 101: Would be good to have a citation to support the "increasing both in frequency and intensity". Otherwise, evidence for this should be shown in the paper.

*Thank you very much for your help, we have added a citation to support this statement (Field et al., 2014), which is the citation provided in the basin plan.*

12. Page 8, line 169: how do the predictions by these 4 models compare with the predictions from the ensemble of models used by the IPCC? With 4 GCMs and 3 emission scenarios, this means 12 climate scenarios.

*We refer to the 3 emission scenarios as "scenarios", while the combination of the 3x4 scenarios and GCMs are termed "forcings" to the GHMs. Thus, we distinguish scenarios proper from modeling outputs (albeit we reckon the emission scenarios are the outcome of IAMs themselves, but they are nonetheless exogenous and typically referred to as scenarios in the climate modeling community).*

*The predictions from the 4 GCMs are taken from ISIMIP and a discussion on these results is available in ISIMIP2b. We do not produce any new outcome here–we are simply describing the inputs used.*

13. Page 11, lines 268-269: This sentence needs rephrasing.

*Following your suggestion, the sentence has been rewritten as follows: "In general, these models include a non-linear component within the objective function, which can be yield or cost.*

14. Page 11, line 270: Need to define variables.

*We have rewritten the sentence to define all the variables. The sentence has been rewritten as follows:" The original parameter, yield ($y_i$) or cost ($c_i$), is replaced by a crop area-dependent function ($y_i = B0_i + B1_i x_i$ or $c_i = \alpha_i + \frac{1}{2}\beta_i x_i$), so that when the area of a crop ($x_i$) expands, its yield decreases (or its cost increases)*

*and vice versa, being $B0_i$, $B1_i$, $\alpha_i$ and $\beta_i$ the calibrating parameters (intercept*

*and slope) for yield and cost linear functions."*

15. Table 1: need to define variables. As far as I can see, only mu_i has been defined.

*We have corrected the table and text and now include definitions.*

16. Figure 3: font size on the axes is too small. At the moment, this plot is not very helpful. Maybe better to give a cumulative frequency (or flow duration) curve?

*The font size of the annexes has been enlarged to make it easier to read.*

17. Caption of Figure 4: averaged across 4 GCMs and 8 GHMs, so an average of 32

sets of model outputs? What is the standard deviation of this set of results?

Estimate of uncertainty in the mean?

*We agree that using best estimates is not the best way to show results in a paper*

*about uncertainty. We nonetheless want to convey the spatial variability of the*

*modeling outcomes. Since this is not a critical result of the model (rather an*

*input), we have removed this figure.*

18. Page 14, line 331: I don't find Figure 5 particularly informative. Can these results be better represented? At the moment, 3 pages of very small figures is not working.

*We now use box-whisker plots to capture uncertainty in three figures, one for each*

*scenario. Note that box-whisker plots quantify uncertainty over the entire basin*

*and do not offer any spatial disaggregation of results. On the other hand, we*

*reckon we cannot show the large number of figures we used in the previous*

*version to present detailed distributed results. Instead, we now show one figure*

*as an example of the potential for the model to produce spatially distributed*

*results, and refer the reader to the appendix D for more detailed information.*

Page 14, line 333: similarly for Figure 6. Need a summarising figure in the paper.

The individual plots can be given in supplementary material, but not in the actual paper.

*See answer to your previous comment.*

19. Page 17, Figure 5: The legend indicates these are Delta values - what is the change with respect to. Is this current profit? If so, over what period?

*This reflects the change with respect to the current observed values (i.e.,*

*profit/employment in the calibration year 2017). We have revised the figure*

*caption to reflect this.*

**Author's changes in the manuscript. Major changes made to the manuscript:**

1. **Definition and Concepts of Uncertainty:**
   - We have expanded the initial section to include a clear definition of uncertainty and the main associated concepts, providing a more solid theoretical framework.

2. **Review of Uncertainty Analysis:**
   - An exhaustive review of uncertainty analysis conducted by several recent authors has been incorporated, enriching the context and relevance of our study.

3. **Box-and-Whisker Plot:**
   - A box-and-whisker plot has been added to better capture and represent the uncertainty in our data, offering a more precise and comprehensible visualization of variability.

4. **Improvement of Figure 5:**
   - Figure 5 has been redesigned to include an example of the model's potential, making the figures larger and easier to interpret. Additionally, we have added a reference to the appendix for readers to access detailed information.

5. **Critical Discussion of Limitations:**
   - The discussion section has been revised to more critically address the limitations of our work and the challenges related to combining intercomparison projects.